# Disrupting the ciliary gradient of active Arl3 affects rod photoreceptor nuclear migration

Amanda M Travis[1], Samiya Manocha[1], Jason R Willer[1], Timothy S Wessler[2], Nikolai P Skiba[3], Jillian N Pearring[1,4]*

[1]Department of Ophthalmology and Visual Science, University of Michigan-Ann Arbor, Ann Arbor, United States; [2]Department of Mathematics, University of North Carolina at Chapel Hill, Chapel Hill, United States; [3]Department of Ophthalmology, Duke University, Durham, United States; [4]Department of Cell and Developmental Biology, University of Michigan–Ann Arbor, Ann Arbor, United States

**Abstract** The small GTPase Arl3 is important for the enrichment of lipidated proteins to primary cilia, including the outer segment of photoreceptors. Human mutations in the small GTPase Arl3 cause both autosomal recessive and dominant inherited retinal dystrophies. We discovered that dominant mutations result in increased active G-protein—Arl3-D67V has constitutive activity and Arl3-Y90C is fast cycling—and their expression in mouse rods resulted in a displaced nuclear phenotype due to an aberrant Arl3-GTP gradient. Using multiple strategies, we go on to show that removing or restoring the Arl3-GTP gradient within the cilium is sufficient to rescue the nuclear migration defect. Together, our results reveal that an Arl3 ciliary gradient is involved in proper positioning of photoreceptor nuclei during retinal development.

*For correspondence:
pearring@umich.edu

Competing interest: The authors declare that no competing interests exist.

## Editor's evaluation

This paper will be of interest to scientists interested in studying ciliogenesis and specifically vertebrate photoreceptors, which are specialized cilia. The study shows that mutations in the small GTP binding protein ARL3 known to cause dominant inherited human retinal dystrophies result in ARL3 hyperactivity, disrupt the normal ciliary gradient of ARL3 activity, and alter nuclear migration affecting retinal development.

## Introduction

Arl3 is a member of the ADP-ribosylation factor (Arf) family of small GTPases that are important for membrane trafficking (*Cavenagh et al., 1994*). Arl3 is ubiquitously expressed in ciliated cells where it regulates the ciliary enrichment of lipidated proteins (*Fansa and Wittinghofer, 2016*). Many prenylated and myristoylated proteins are shuttled by the Arl3 effectors PDEδ and UNC119A/B, respectively (*Fansa et al., 2016*; *Zhang et al., 2011*; *Zhang et al., 2004*). PDEδ and UNC119A/B are chaperones that sequester the lipid anchor moiety of proteins within an intramolecular cavity to enable detachment of these proteins from membranes (*Hanzal-Bayer et al., 2002*). Arl3 is specifically activated in the cilium by its guanine exchange factor (GEF) Arl13B and co-GEF BART (*ElMaghloob et al., 2021*; *Gotthardt et al., 2015*). Active GTP-bound Arl3 then binds to the chaperones allowing them to unload their lipidated cargo at the cilium. Arl3 is inactive outside the cilium as it is inactivated by RP2, its GTPase activating protein (GAP) (*Veltel et al., 2008a*; *Veltel et al., 2008b*) that is found enriched at the base of the cilium, although it is also present at other locations in the cell (*Evans et al.,*

2010; *Holopainen et al., 2010*). Germline Arl3 knockout mice have phenotypes generally associated with human ciliopathies, including polycystic kidneys and photoreceptor degeneration (*Schrick et al., 2006*).

In photoreceptors, Arl3 regulates enrichment of many lipidated proteins essential for eliciting the visual response within the outer segment, a modified primary cilium (reviewed in *Frederick et al., 2020*). Recently, human mutations in Arl3 have been linked to various forms of retinal degeneration: R99I causes autosomal recessive cone-rod dystrophy (*Sheikh et al., 2019*), R149H or R149C cause recessive Joubert syndrome (*Alkanderi et al., 2018*), compound heterozygous T31A/C118F causes rod-cone dystrophy (*Fu et al., 2021*), Y90C causes autosomal dominant retinitis pigmentosa (*Holtan et al., 2019*; *Strom et al., 2016*), and D67V causes autosomal dominant retinal degeneration (*Ratnapriya et al., 2021*). The R149H/C mutation is present at the Arl3–Arl13B interface and was shown to prevent Arl3 activation by Arl13B, which causes reduced enrichment of lipidated proteins in the cilium (*Alkanderi et al., 2018*). Additionally, the Arl3 mutations C118F, Y90C, and D67V have also been predicted to impair its interactions with binding partners (*Fu et al., 2021*; *Ratnapriya et al., 2021*; *Strom et al., 2016*); while the R99I, T31A, C118F, and Y90C mutations are predicted to destabilize Arl3 (*Fu et al., 2021*; *Holtan et al., 2019*; *Sheikh et al., 2019*). These studies predict that human disease is linked to a loss of Arl3 function; however, the fact that Arl3 mutations can present as dominant retinal dystrophy suggests that some of these variants result in excessive Arl3 activity.

Studies from mice support this idea. Both a rod-specific Arl3 knockout mouse or a transgenic mouse overexpressing constitutively active Arl3-Q71L in rods show severe retinal degeneration by 2 months and result in measurable effects on lipidated protein delivery to the outer segment (*Hanke-Gogokhia et al., 2016*; *Wright et al., 2016*). However, only mice with Arl3-Q71L transgenic expression in rods had a subset of rod nuclei displaced from the outer nuclear layer (ONL) into the inner nuclear layer (INL) (*Wright et al., 2016*). Improper rod nuclear displacement into the INL has also been observed in a RP2 knockout mouse (*Li et al., 2013*; *Mookherjee et al., 2015*), further suggesting that disrupting Arl3 activity in rods could lead to a nuclear migration phenotype.

Retinal neurons are organized into three distinct nuclear layers (listed from apical to basal): the ONL, the INL, and the ganglion cell layer (GCL). During development, all retinal neurons are born apically and then migrate to their final retinal layer where they elaborate and make connections (*Baye and Link, 2008*). For cones, whose nuclei are arranged at the apical side of the ONL, it is known that their nuclear position can be influenced by either knocking out RASGRF2 the GEF for the Ras/Rho/Rac family of small GTPases (*Jimeno et al., 2016*), altering dopamine signaling through the D4 receptor (*Tufford et al., 2018*), or uncoupling the nucleus from microtubules (*Yu et al., 2011*; *Razafsky et al., 2012*). Less is known about how rod nuclear position is determined, but microtubule coupling to the nucleus is important. Mouse models disrupting this connection have found rod nuclei mislocalized to the INL (*Yu et al., 2011*) and, more recently, a study found that rods undergo rapid, dynein-dependent apical nuclear translocations during the formation of the retinal nuclear layers between P0 and P8 (*Aghaizu et al., 2021*). Exactly how constitutively active Arl3 might impair the migration of rod nuclei has not been investigated.

We hypothesized that the autosomal dominant human Arl3 mutations are causing constitutively active Arl3 and might result in a nuclear migration defect. Using in vitro biochemistry and cell-based experiments, we found that dominant Arl3 mutations, D67V and Y90C, result in increased cellular activity through distinct changes to their GTPase function. D67V is constitutively active and Y90C is fast cycling. Using in vivo mouse experiments, we further show that this aberrant activity disrupts the Arl3-GTP gradient in the cilium causing defects in rod nuclear migration. Importantly, we can rescue the Arl3-Y90C migration phenotype by either removing the Arl3-GTP gradient entirely or restoring active Arl3-GTP to the cilium highlighting that the cilium is important for proper photoreceptor migration during retinal development.

## Results

### Expression of dominant Arl3 mutants in rod photoreceptors causes nuclear mislocalization

We investigated whether the dominantly inherited human mutations of Arl3 caused a nuclear migration phenotype by expressing exogenous FLAG-tagged human Arl3-D67V and Arl3-Y90C in mouse

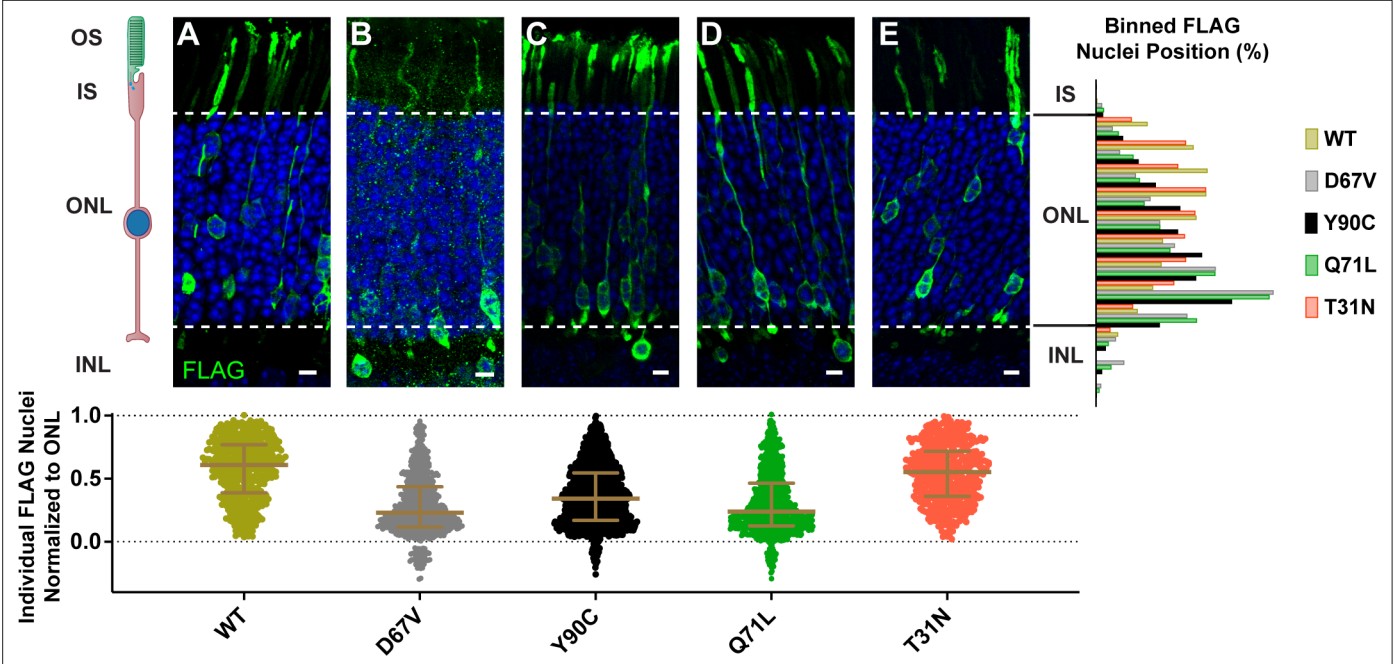

**Figure 1.** Expression of dominant Arl3 mutants in rod photoreceptors causes nuclear mislocalization. Representative images showing cross-sections through wild-type mouse retinas electroporated with Arl3-3XFLAG (**A**), Arl3-D67V-FLAG (**B**), Arl3-Y90C-FLAG (**C**), Arl3-Q71L-FLAG (**D**), or Arl3-T31N-FLAG (**E**) and immunostained with anti-FLAG antibodies (green). Nuclei are counterstained with Hoechst (blue). Scale bars, 5 µm. Here and in all subsequent figures the nuclear position of electroporated rods is represented as follows: Below each image, a scatter plot shows the location of every Arl3-FLAG-expressing nuclei, normalized to its position within the ONL with the apical edge set to 1.0 and the basal edge set to 0.0 (dashed lines in the images and graphs). Median and interquartile range are indicated in brown. To the right, a histogram shows the relative distribution of the nuclear position for each construct with nuclei sorted into 10 ONL and 3 INL bins. A minimum of three expressing eyes were analyzed for each construct. Abbreviations: outer segments (OS), inner segments (IS), outer nuclear layer (ONL), and inner nuclear layer (INL).

rods using in vivo electroporation. FLAG immunostaining of retinal cross-sections showed wild-type Arl3-FLAG present throughout the rod photoreceptor including, notably, within the outer segment at P21 (*Figure 1A*). Both Arl3-D67V-FLAG and Arl3-Y90C-FLAG expression and localization were similar to wild-type Arl3-FLAG, with perhaps a trend toward less Arl3-D67V within the outer segments although this was not quantified. Strikingly, though, many of the Arl3-D67V and Arl3-Y90C nuclei were mislocalized to the INL (*Figure 1B, C*), 7.4% and 2.9% of nuclei, respectively. This nuclear displacement is similar to transgenic expression of the GTP-locked mutant Arl3-Q71L (*Wright et al., 2016*) and our own electroporation of Arl3-Q71L (*Figure 1D*), in which 5.4% of nuclei were mislocalized to the INL. Additionally, even the Arl3-D67V and Arl3-Y90C nuclei localized within the ONL are predominantly displaced basally, while the nuclei of wild-type Arl3-FLAG rods are distributed more evenly throughout the ONL. To quantify this shift, the skewness of the Gaussian distribution of FLAG-expressing nuclei within the ONL was compared between Arl3 mutants and wild-type Arl3. A skewness of zero describes a normal symmetrical distribution, while negative or positive skewness values describe a shift in the distribution apically or basally within the ONL, respectively. All the mutants that had rod nuclei mislocalized to the INL—D67V, Q71L, and Y90C—were positively skewed and statistically significant from wild-type (*Table 1*). This shift in rod nuclear position was not seen with overexpression of a GDP-bound inactive mimetic, Arl3-T31N-FLAG, suggesting that increased Arl3 activity during development causes a defect in rod nuclear migration (*Linari et al., 1999*; *Figure 1E*).

## Arl3-Y90C is a fast cycling GTPase

Because Arl3-D67V and Arl3-Y90C phenocopied the Arl3-Q71L mislocalization of rod nuclei, we hypothesized that these mutations would also result in constitutively active Arl3. We first tested whether the Arl3 mutations are GTPase impaired by assessing their ability to interact with the Arl3 effector PDEδ. A GST-PDEδ pulldown was performed from AD-293 cell lysates expressing FLAG-tagged Arl3 mutants. As expected, inactive Arl3-T31N did not bind to GST-PDEδ while the constitutively active

**Table 1.** Summary of Arl3 mutants.
Human retinal phenotypes, GTPase function, mouse rod nuclear migration defect, and statistics are detailed with references listed.

| Arl3 mutant | Co-expression | Neuronal migration defect | % INL | Skew | p value | Nucleotide-binding properties | Ref | Human disease | Ref |
|---|---|---|---|---|---|---|---|---|---|
| Wild-type | | No | 0 | −0.2232 | | GEF Arl13B/GAP RP2 | *Gotthardt et al., 2015*; *Veltel et al., 2008b* | | |
| T31A | | Unknown | | | | Unknown | | Compound RD | *Fu et al., 2021* |
| T31N | | No | 0 | −0.1866 | 0.9997 | No GTP binding | *Linari et al., 1999* | n/a | |
| T31N/Y90C | | No | 0 | −0.4093 | 0.8193 | No GTP binding | This paper | n/a | |
| D67V | | Yes | 7.37 | 0.5818 | <0.0001 | Active w/ select effector binding | This paper | Dominant RD | *Ratnapriya et al., 2021* |
| Q71L | | Yes | 5.42 | 0.6195 | <0.0001 | No GTP hydrolysis | *Linari et al., 1999*; *Veltel et al., 2008b* | n/a | |
| | +UNC119 | Yes | 3.46 | 0.8616 | <0.0001 | | | | |
| | +PDEδ | Yes | 8.31 | 0.4449 | 0.0037 | | | | |
| pRK-Q71L | | Yes | 3.68 | 0.5328 | <0.0001 | No GTP hydrolysis | *Linari et al., 1999*; *Veltel et al., 2008b* | n/a | |
| | +PDEδ | No | 0 | 0.1224 | 0.4546 | | | | |
| Y90C | | Yes | 2.89 | 0.3053 | 0.0002 | Fast cycling | This paper | Dominant RD | *Strom et al., 2016*; *Holtan et al., 2019* |
| | +Arl13B | No | 0 | 0.1108 | 0.1276 | | | | |
| | +UNC119 | No | 0 | −0.0898 | 0.995 | | | | |
| | +PDEδ | No | 0.26 | −0.4235 | 0.3665 | | | | |
| | +NPHP3 | No | 0.84 | −0.2929 | 0.9994 | | | | |
| | +INPP5E | No | 0 | −0.0367 | 0.7673 | | | | |
| | +INPP5E-C644A | Yes | 4.22 | 0.2357 | 0.0055 | | | | |
| | +Rnd1 | Yes | 2.94 | 0.5827 | <0.0001 | | | | |
| Y90C/R149H | | No | 0.71 | 0.0987 | 0.0921 | Fast cycling | This paper | n/a | |
| R99I | | Unknown | | | | Unknown | | Recessive RD | *Sheikh et al., 2019* |
| C118F | | Unknown | | | | Unknown | | Compound RD | *Fu et al., 2021* |
| D129N | | Yes | 3.67 | 0.2855 | 0.0113 | Fast cycling | *Gotthardt et al., 2015* | n/a | |
| R149H | | No | 0 | 0.1229 | 0.051 | No GEF binding | *Alkanderi et al., 2018* | Recessive RD | *Alkanderi et al., 2018* |

Arl3-Q71L is bound. Arl3-D67V was also pulled down by GST-PDEδ, indicating that it is constitutively active. Surprisingly, however, Arl3-Y90C did not bind to GST-PDEδ, indicating that it is not constitutively active (*Figure 2A*).

To assess Arl3-Y90C's capacity to bind GTP we performed the GST-PDEδ pulldown under several different conditions. We attempted to load Arl3-Y90C with GTP by incubating Arl3-Y90C-FLAG-expressing AD-293 cell lysates with EDTA (to imitate the function of the GEF by chelating out the nucleotide coordinating $Mg^{2+}$) and found that, if there is excess GTP in the buffer, Arl3-Y90C can bind GTP as measured by GST-PDEδ binding, similar to both endogenous Arl3 (as seen below in the same blot) and Arl3-FLAG (*Figure 2—figure supplement 1*). In fact, Arl3-Y90C binds GTP under these nucleotide exchange promoting conditions more efficiently than the endogenous Arl3 expressed by these cells. Interestingly, when excess GTP was added Arl3-Y90C also underwent nucleotide exchange in the presence of $Mg^{2+}$, which blocks nucleotide exchange for endogenous Arl3 (*Figure 2B*) and

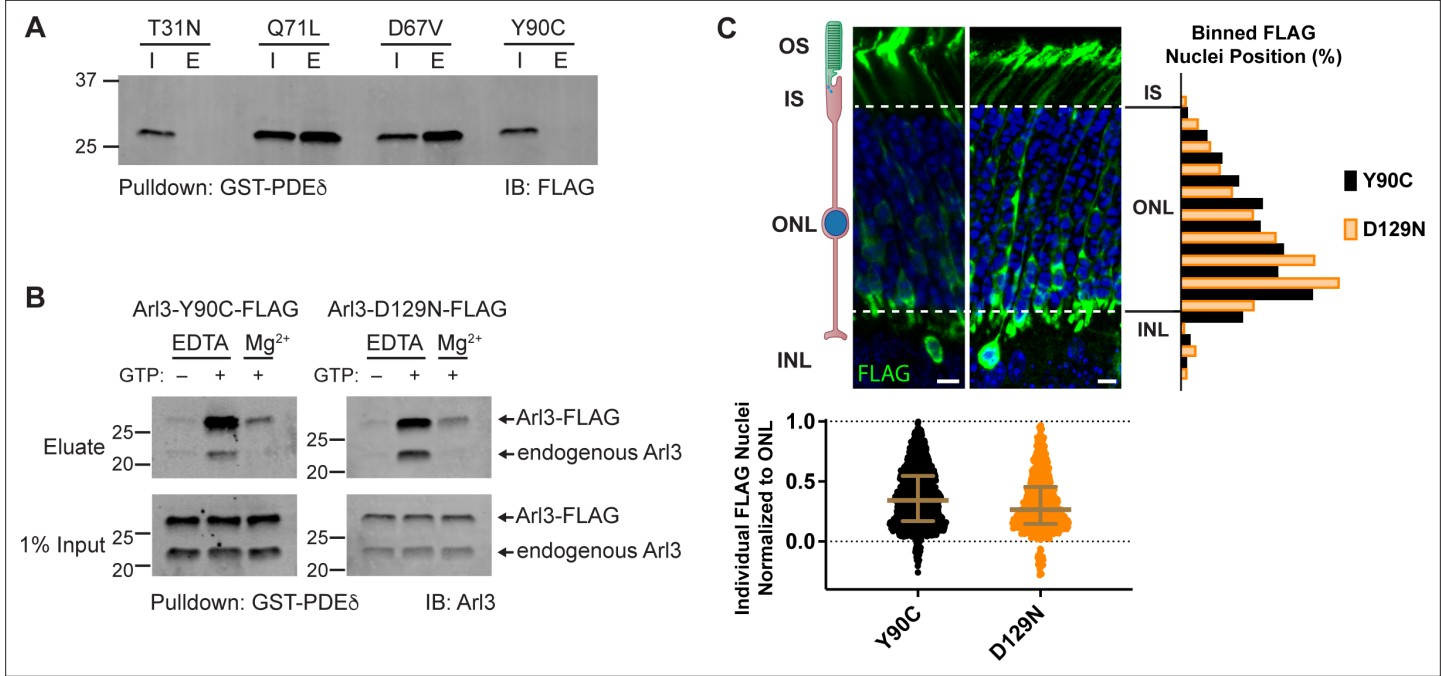

**Figure 2.** Arl3-Y90C is a fast cycling GTPase. (**A**) 1% input (I) and eluates (E) from GST-PDEδ pulldowns using AD-293 lysates expressing 3XFLAG-tagged Arl3 mutants immunoblotted with anti-FLAG antibodies. (**B**) Arl3-Y90C-FLAG or Arl3-D129N-FLAG lysates were incubated with 10 mM ethylenediaminetetraacetic acid (EDTA) and/or 10 mM GTP, spiked with Mg²⁺, and precipitated with GST-PDEδ. Westerns immunoblotted for Arl3 with eluates shown on top, and 1% inputs shown below. Arl3-FLAG and endogenous Arl3 bands labeled. (**C**) Representative retinal cross-sections from Arl3-Y90C-FLAG- or Arl3-D129N-expressing rod photoreceptors stained with anti-FLAG antibodies (green) and counterstained with Hoechst (blue). Scale bars, 5 μm. Nuclear position of electroporated rods represented as described in *Figure 1*.

The online version of this article includes the following source data and figure supplement(s) for figure 2:

**Source data 1.** Raw western blot images.

**Figure supplement 1.** GTP loading of wild-type Arl3-FLAG.

**Figure supplement 2.** Unlike Arl3-Q71L, Arl3-D67V does not form a ternary complex with RP2 and PDEδ.

Arl3-FLAG (*Figure 2—figure supplement 1*). This result indicates possible GEF-independent nucleotide exchange in Arl3-Y90C, a behavior that is often described as fast cycling of a small GTPase. To confirm, we tested the behavior of Arl3-D129N (*Gotthardt et al., 2015*), a homologous mutant of the fast cycling Ras-D119N (*Cool et al., 1999*), in this assay and found it behaved biochemically identical to Arl3-Y90C (*Figure 2B*). To confirm that this biochemical result reveals small GTPase behavior relevant to the migration phenotype, we electroporated Arl3-D129N-FLAG in mouse rod photoreceptors and found that the fast cycling Arl3-D129N replicated the Arl3-Y90C migration phenotype with 3.7% of Arl3-D129N-FLAG-expressing cells mislocalized to the INL (*Figure 2C*). Together, our data suggest that the Arl3-Y90C mutation results in fast cycling behavior that causes excess Arl3 activation in vivo.

## Arl3-Y90C behaves as a dominant negative

Another key component of fast cycling GTPase behavior is increased binding to the GEF due to an increased nucleotide-free state (*Cool et al., 1999*; *Gotthardt et al., 2015*). We tested whether the Arl3-Y90C mutation also caused increased binding to Arl13B, the ciliary Arl3 GEF, by analyzing the ability of FLAG-tagged Arl3 mutants to co-immunoprecipitate GFP-tagged Arl13B from cell culture lysates. As controls we used the fast cycling D129N mutation previously shown to bind Arl13B in yeast-2-hybrid experiments (*Gotthardt et al., 2015*) and the Arl3-R149H mutant previously shown to disrupt interaction with Arl13B (*Alkanderi et al., 2018*). In *Figure 3A*, western blot analysis of the FLAG eluates using an anti-GFP antibody shows Arl3-Y90C binds Arl13B. When we introduced the R149H mutation to the Arl3-Y90C mutant, Arl13B binding was disrupted similarly to the Arl3-R149H single mutant (*Figure 3A*), confirming that Arl3-Y90C binding to Arl13B is at the same interface as wild-type binding. This result suggests that increased Arl13B binding to the Arl3-Y90C mutant is

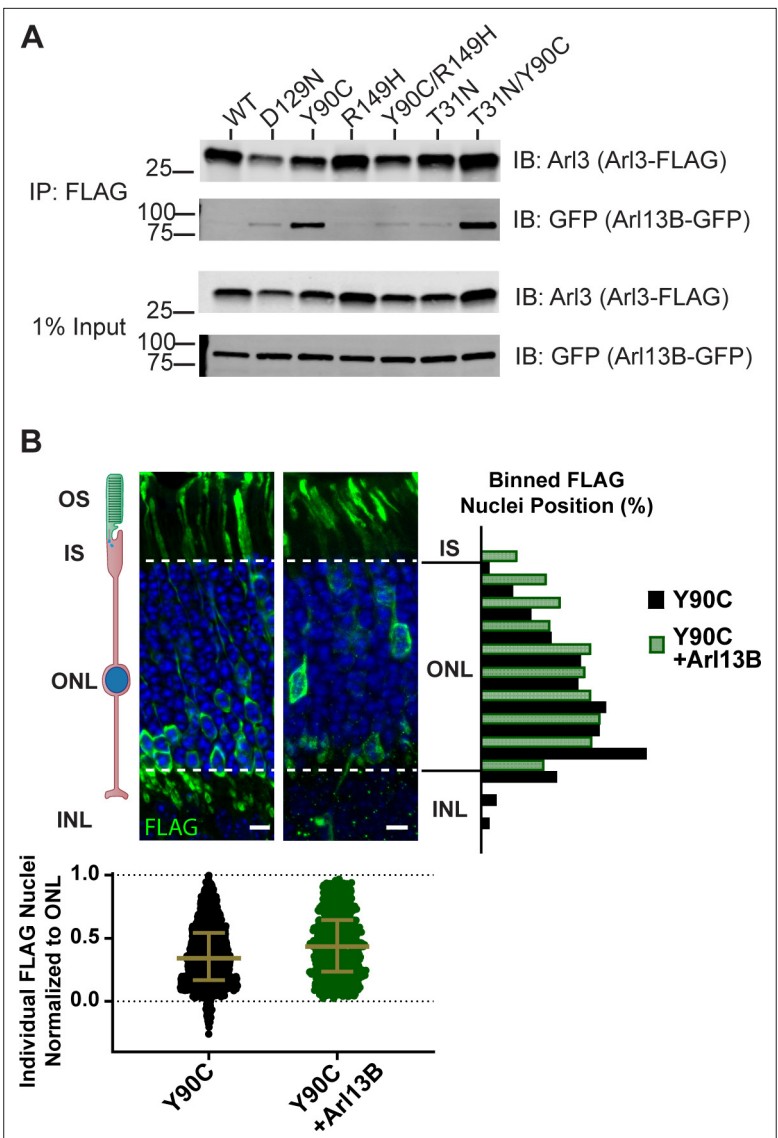

**Figure 3.** Arl3-Y90C acts as a dominant negative. (**A**) FLAG immunoprecipitation from AD-293 lysates expressing Arl13B-GFP and Arl3-FLAG mutants. Westerns immunoblotted for Arl3 and GFP with 1% inputs shown on top and FLAG eluates on bottom. (**B**) Representative retinal cross-sections from Arl3-Y90C-FLAG alone or co-expressed with Arl13B-myc and rods stained with anti-FLAG antibodies (green) and counterstained with Hoechst (blue). Scale bars, 5 µm. Nuclear position of electroporated rods represented as described in *Figure 1*.

The online version of this article includes the following source data and figure supplement(s) for figure 3:

**Source data 1.** Raw western blot images.

**Figure supplement 1.** Arl3-Y90C overexpression in rods does not alter endogenous Arl13B localization.

**Figure supplement 2.** Arl13B-myc localization is not changed by co-expression with Arl3-Y90C-FLAG.

**Figure supplement 3.** Analyzing the amount of Arl13B-GFP bound to Arl3-FLAG mutants expressed in AD-293 cells.

due to an increased nucleotide-free state of Arl3-Y90C rather than fundamental alterations within its Arl13B-binding site, a conclusion supported by the continued binding of Arl3-Y90C to Arl13B after an additional T31N mutation, which decouples GTP-binding state from Arl13B-binding behavior of Arl3-Y90C.

Importantly, despite the data showing strong Arl3-Y90C binding to Arl13B, endogenous Arl13B remains predominantly localized to the outer segment and does not mislocalize in rods expressing

Arl3-Y90C-FLAG (*Figure 3—figure supplement 1*), likely due to the palmitoylation of Arl13B (*Roy et al., 2017*). Further, electroporation of Arl13B-myc in rods results in myc staining primarily in the outer segment with some present in the cell body and synapse. The same Arl13B localization pattern was observed when Arl13B-myc was co-expressed with Arl3-Y90C confirming that Arl3-Y90C binding does not alter Arl13B ciliary localization (*Figure 3—figure supplement 2*).

Although expression of Arl3-Y90C does not cause mislocalization of Arl13B, Arl3-Y90C could have profound effects as a dominant negative by binding to Arl13B—Arl13B sequestered by Arl3-Y90C would no longer activate endogenous Arl3. If Arl3-Y90C is causing the migration defect because it is acting as a dominant negative, migration should be rescued if enough Arl13B is present in the outer segment to activate endogenous Arl3 and therefore overcome the dominant negative consequences of Arl3-Y90C expression. To test this, the localization of rod nuclei of cells expressing both Arl3-Y90C-FLAG and Arl13B-MYC was examined. This cell population no longer has the migration defect observed with only Arl3-Y90C-FLAG expression indicating that the migration defect is caused at least in part by Arl3-Y90C acting as a dominant negative (*Figure 3B*).

## Arl3-Y90C undergoes GEF-independent activation in cell culture

The fact that Arl3-Q71L and Arl3-D67V, which are both constitutively active, also cause the migration defect while it is not seen in Arl3-KO mice argues that the dominant negative role of Arl3-Y90C is not a sufficient explanation for this phenotype, so we next examined the nucleotide status of Arl3-Y90C in the cell. Since the fast cycling nature of Arl3-Y90C precludes any measurement of the biological nucleotide-binding state using GST-PDEδ pulldowns, we incubated AD-293 cells expressing wild-type or mutant FLAG-tagged Arl3 constructs with the permeable crosslinker DSS and then performed FLAG immunoprecipitations to identify Arl3 effector complexes by western blot using an anti-Arl3 antibody. As proof of concept, we performed this in vivo crosslinking for the inactive Arl3-T31N and constitutively active Arl3-Q71L mutants. When DSS crosslinker was added, three major Arl3 complexes were visible in the presence of active Arl3-Q71L (*Figure 4A*). Mass spectrometry analysis of DSS treated lysates identified three known Arl3-binding partners: RP2, UNC119, and BART in the Arl3-Q71L but not in the Arl3-T31N sample (*Figure 4B*).

It was previously shown that a double mutation in Arl3, E164A/D168A, reduces affinity for RP2 (*Veltel et al., 2008a*). To confirm that RP2 is forming the ~65 kDa complex with Arl3-Q71L in the in vivo crosslinking experiments, we generated and tested a Arl3-Q71L/E164A/D168A triple mutant (Q71L[noRP2]) that maintains active Arl3 binding to effectors but not RP2 (*Figure 4C*, lane 1). Western blot using antibodies against RP2 confirmed that the ~65 kDa band is indeed a complex between RP2 and Arl3-Q71L (*Figure 4C*, lane 2). We took a similar approach to confirm that UNC119 is forming the ~55 kDa complex with Arl3-Q71L by using Arl3-D67V. This mutant has constitutive activity in our GST-PDEδ in vitro pulldowns (*Figure 2A*) but is unable to interact with UNC119A-myc (*Figure 4—figure supplement 1*). Indeed, the ~55 kDa band is strongly reduced in the presence of Arl3-D67V, confirming that this complex is between Arl3 and UNC119A (*Figure 4C*, lane 3). We believe the final complex at ~45 kDa in each constitutively active condition is between Arl3-GTP and BART but were unable to directly verify. Overall, we show that the in vivo crosslinking experiments measure the presence of Arl3-GTP in a transfected populations of cells via effector binding. As an aside, we also found that in vivo crosslinking can reveal differences in how Arl3 mutants bind effectors since covalent crosslinking of secondary structures can result in proteins running faster in the gel (e.g., the Arl3 band runs faster than 25 kDa after exposure to the crosslinker, *Figure 4A*). In *Figure 4C*, the red arrowhead highlights a FLAG- and RP2-positive band in the presence of Arl3-D67V that runs faster than the same complex found in the Arl3-Q71L lane. This likely indicates differences in RP2 binding between the two constitutively active mutants that result in different crosslinking-site availability.

We then performed the in vivo crosslinking on cells expressing Arl3-Y90C-FLAG and found that Arl3-Y90C formed complexes with UNC119 and BART but did not make a stable complex with RP2 (*Figure 4D*). This result suggests that Arl3-Y90C is active in AD-293 cells and binding effectors but, unlike Arl3-Q71L, is inactivated by RP2 and therefore does not form a stable complex with RP2. The fast cycling mutant Arl3-D129N showed the same pattern of Arl3 complex formation. To understand the relative level of active Arl3-Y90C in cells, we measured the band intensity of each Arl3 in complex with UNC119 or BART and normalized to the same bands in Arl3-Q71L. We found that both fast cycling mutants, Arl3-D129N and Arl3-Y90C, had significantly higher levels than wild-type Arl3

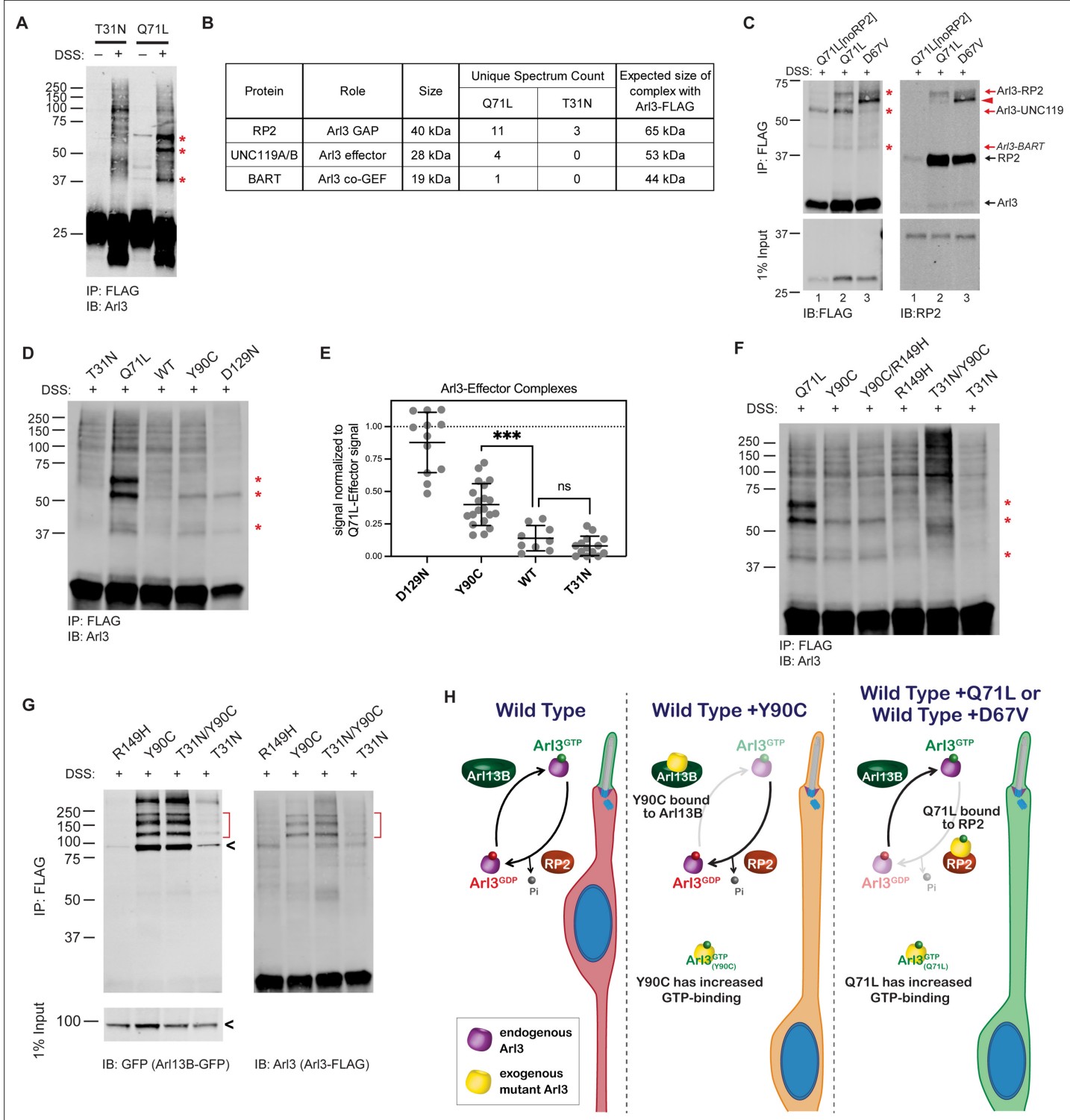

**Figure 4.** Arl3-Y90C undergoes guanine exchange factor (GEF)-independent activation in cell culture. (**A**) FLAG immunoprecipitation from DSS-crosslinked or non-crosslinked AD-293 cell lysates expressing either Arl3-T31N-FLAG or Arl3-Q71L-FLAG immunoblotted with anti-Arl3 antibodies. Red asterisks indicate the three prominent bands that appear in crosslinked cells expressing Arl3-Q71L-FLAG. (**B**) Table displaying the Arl3-binding partners identified by mass spectroscopy of FLAG immunoprecipitations from either Arl3-Q71L-FLAG or Arl3-T31N-FLAG lysates after DSS crosslinking. (**C**) Crosslinked FLAG immunoprecipitation from either Arl3-Q71L/E164A/D168A-FLAG (Q71L[noRP2]), Arl3-Q71L-FLAG, or Arl3-D67V-FLAG probed with either anti-FLAG or anti-RP2 antibodies. Red asterisks and red arrows indicate crosslinked Arl3 complexes. Arl3-FLAG complexed with RP2 is found in two bands, the lower band indicated with red arrowhead. Black arrows indicate the non-crosslinked proteins. (**D**) Representative western blot showing

*Figure 4 continued on next page*

*Figure 4 continued*

crosslinked FLAG immunoprecipitation from Arl3-FLAG mutants: T31N, Q71L, Y90C, and D129N. Red asterisks indicate the three crosslinked Arl3 complexes identified from Arl3-Q71L. (**E**) The signal intensity of crosslinked Arl3 complexes for each mutant normalized to the intensity of Arl3-Q71L complexes on the same blot. ns, p = 0.8108 and ***p = 0.0007. (**F**) Crosslinked FLAG immunoprecipitation from Arl3-FLAG double mutants and controls. Red asterisks indicate the three bands that appear in the presence of Arl3-Q71L. (**G**) Crosslinked FLAG immunoprecipitation from lysates expressing both Arl13B-GFP and Arl3-FLAG mutants. The red brackets identify Arl13B-Arl3 complexes and the black arrowheads non-crosslinked Arl13B-GFP. (**H**) Cartoon model depicts Arl3 GTPase cycle in 'immature' rod photoreceptors under wild-type conditions and how it is disrupted in the presence of exogenous Arl3-Y90C, Arl3-D67V, or Arl3-Q71L. Red indicates the presence of inactive Arl3-GDP, green indicates the presence of active Arl3-GTP, and orange indicates a mixed population of Arl3-GDP/Arl3-GTP.

The online version of this article includes the following source data and figure supplement(s) for figure 4:

**Source data 1.** Excel table with mass spectrometry data from Arl3-Q71L and Arl3-T31N crosslinking experiments.

**Source data 2.** Raw western blot images.

**Figure supplement 1.** Unlike Arl3-Q71L, Arl3-D67V does not bind UNC119A.

(*Figure 4E*). The relative amounts of effector complex formation correlate to migration phenotype: wild-type has normal migration and is not significantly different than Arl3-T31N while the amount of Arl3-Y90C and Arl3-D129N activation is closer to Arl3-Q71L, resulting in defective migration.

We also performed the in vivo crosslinking on cells expressing the Arl3-Y90C/R149H-FLAG double mutant construct to verify whether the aberrant activity of the fast cycling Y90C mutation is maintained in the absence of GEF binding. In *Figure 4F*, we see increased expression of the Arl3-UNC119 and Arl3-BART complexes in the Y90C/R149H double mutant compared to the R149H control. Importantly, the Y90C mutation no longer forms effector complexes in the presence of the T31N mutation, so binding of GTP is indeed still necessary for activity in the presence of the Y90C mutation. Together, these results confirm that GEF-independent nucleotide exchange to GTP is activating the Arl3-Y90C mutant in cells.

Our previous FLAG immunoprecipitation experiments confirmed that Arl3-Y90C has tight binding to Arl13B (*Figure 3A*), but to test whether this is occurring in the cell we performed in vivo crosslinking experiments in the presence of Arl13B-GFP. As a control, we used Arl3-R149H-FLAG and found that it was unable to form complexes with expressed Arl13B-GFP as expected (*Figure 4G*). In contrast, in vivo crosslinking captured complexes between Arl13B-GFP and Arl3-Y90C showing that GEF binding is a feature of Arl3-Y90C in cells. We also found Arl13B-GFP complexes with Arl3-T31N/Y90C, but not with Arl3-T31N, suggesting that the Arl3-Y90C ability to tightly bind the GEF does not depend on its nucleotide status (*Figure 4G*). Together our biochemical data show that Arl3-Y90C and Arl3-Q71L/Arl3-D67V both have aberrant activity and form stable complexes with important members of the Arl3 GTPase cycle, Arl13B and RP2, respectively. *Figure 4H* shows a cartoon model of a developing 'immature' rod photoreceptor and how Arl3 activity effector binding is altered in the presence of different dominant mutations.

## Aberrant Arl3 activity drives the rod nuclear migration phenotype

We hypothesized that the nuclear migration defect requires aberrant activation of Arl3 within rods. If this is true, then we would expect that eliminating all Arl3 activity would allow for normal nuclear migration. We already found that the Arl3-T31N/Y90C double mutant is inactive within the cell (*Figure 4F*) but remains tightly bound to Arl13B (*Figure 3A*), which we predict would prevent activation of endogenous Arl3 in rods. To determine whether wild-type Arl3 can be activated by Arl13B in the presence of Arl3-Y90C/T31N we performed FLAG immunoprecipitations after in vivo crosslinking in AD-293 cell co-expressing Arl13B-GFP with wild-type Arl3-FLAG and mutant Arl3-myc. We then assessed in vivo effector binding of the wild-type Arl3-FLAG by western blot. We found that wild-type Arl3-FLAG is activated by Arl13B-GFP shown by the presence of the UNC119 and BART complexes (*Figure 5A*). However, in the presence of myc-tagged Arl3-Y90C or Arl3-T31N/Y90C, the wild-type Arl3-FLAG no longer forms these effector complexes (*Figure 5A*). The absence of these bands is due to the Arl3-Y90C binding to Arl13B-GFP, as we see normal wild-type Arl3-FLAG activation in the presence of Arl3-T31N-myc.

These results show that wild-type Arl3 is unable to be activated in the presence of Arl3-Y90C, confirming our working model from *Figure 4H*. We then expressed the Arl3-T31N/Y90C-FLAG construct in mouse rod photoreceptors to test the migration phenotype in the absence of endogenous

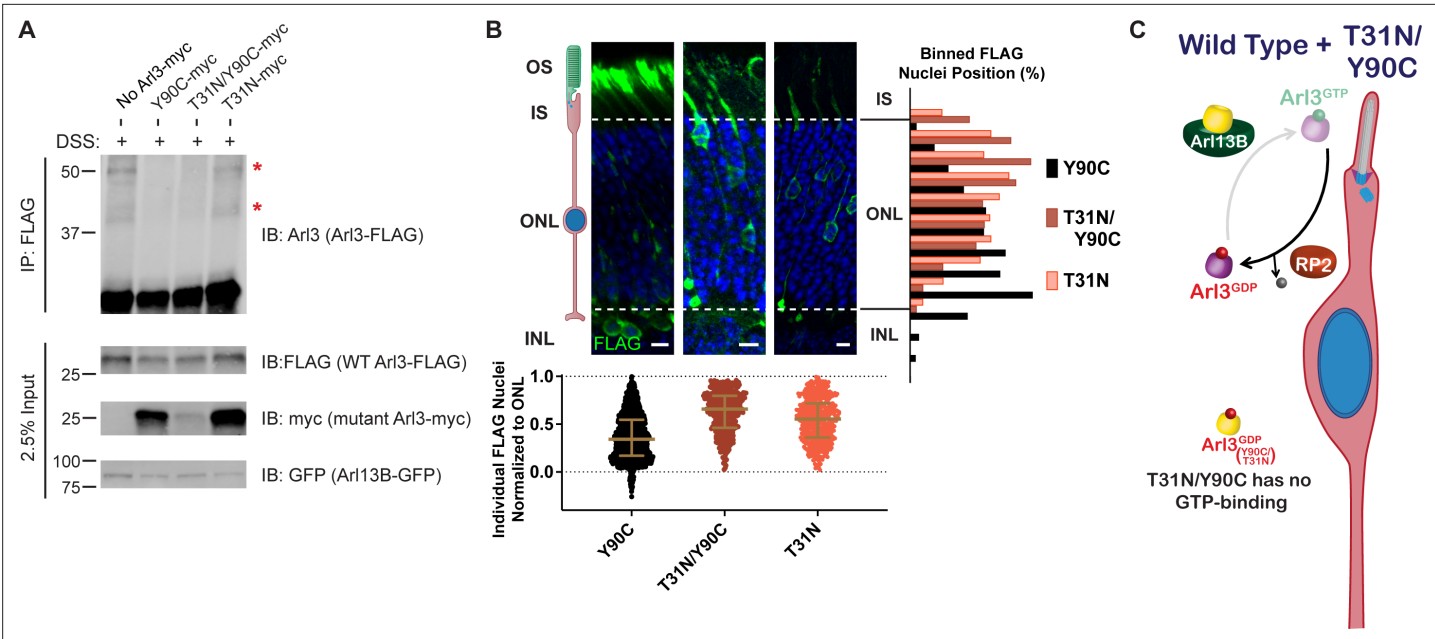

**Figure 5.** Aberrant Arl3 activity drives the rod nuclear migration phenotype. (**A**) Crosslinked FLAG immunoprecipitation from AD-293 lysates expressing Arl13B-GFP, wild-type Arl3-FLAG, and myc-tagged Arl3 mutants. Red asterisks indicate the two crosslinked Arl3 complexes that appear in the presence of Arl3-Y90C. (**B**) Representative retinal cross-sections from Arl3-Y90C-FLAG-, Arl3-T31N/Y90C-FLAG-, or Arl3-T31N-FLAG-expressing rod photoreceptors stained with anti-FLAG antibodies (green) and Hoechst (blue). Scale bars, 5 μm. Nuclear position of electroporated rods represented as described in *Figure 1*. (**C**) Cartoon model portraying disrupted Arl3 GTPase cycle in 'immature' rod photoreceptors in the presence of exogenous Arl3-T31N/Y90C mutant (yellow). Red indicates Arl3 is inactive throughout the rod and nuclear positioning is normal.

The online version of this article includes the following source data for figure 5:

**Source data 1.** Raw western blot images.

Arl3 activation and aberrant Arl3 activity, essentially mimicking a loss of function phenotype. We found that Arl3-T31N/Y90C-FLAG did not result in a nuclear migration defect (*Figure 5B*), suggesting that Arl3 activity is driving the disruption of nuclear migration in rods (*Figure 5C*).

## Increase in chaperones for lipidated proteins can rescue the Arl3-Y90C rod migration defect

Given that elevated levels of Arl3 activity within the cell lead to the migration defect, we predicted that we could rescue the phenotype by increasing the pool of available effectors, essentially chelating the excess active Arl3-Y90C and/or the Arl3-Y90C bound to Arl13B. In cell culture, we find that expression of either UNC119-myc or Arl13B-GFP decreases the formation of active complexes between Arl3-Y90C-FLAG and endogenous effectors (*Figure 6—figure supplement 1*). We tested this hypothesis in vivo by overexpressing the chaperones, UNC119 and PDEδ, with Arl3-Y90C-FLAG in mouse rods and found that rod nuclei were normally distributed throughout the ONL (*Figure 6A*). Expression of these Arl3 effectors likely sequesters active Arl3-Y90C to allow for both Arl13B-dependent activation of endogenous Arl3 and chaperone participation in lipidated protein transport (*Figure 6B*).

Surprisingly, when UNC119 or PDEδ were co-expressed with Arl3-Q71L we found that rod nuclei were still mislocalized (*Figure 6C*). We previously observed that the level of Arl3-Q71L activity within AD-293 cells was much greater than Arl3-Y90C (*Figure 4D, E*), so to rescue Arl3-Q71L we attempted to better match the activity level of Arl3-Y90C by switching Arl3-Q71L-FLAG from the highly expressing rhodopsin promoter to a lower expressing rhodopsin kinase promoter (*Khani et al., 2007*). The lower expression of Arl3-Q71L-FLAG still resulted in a rod migration defect; however, co-expression of PDEδ was now able to rescue the phenotype (*Figure 6C*). These data indicate that excess activation beyond a given threshold leads to an impairment in proper rod nuclear migration.

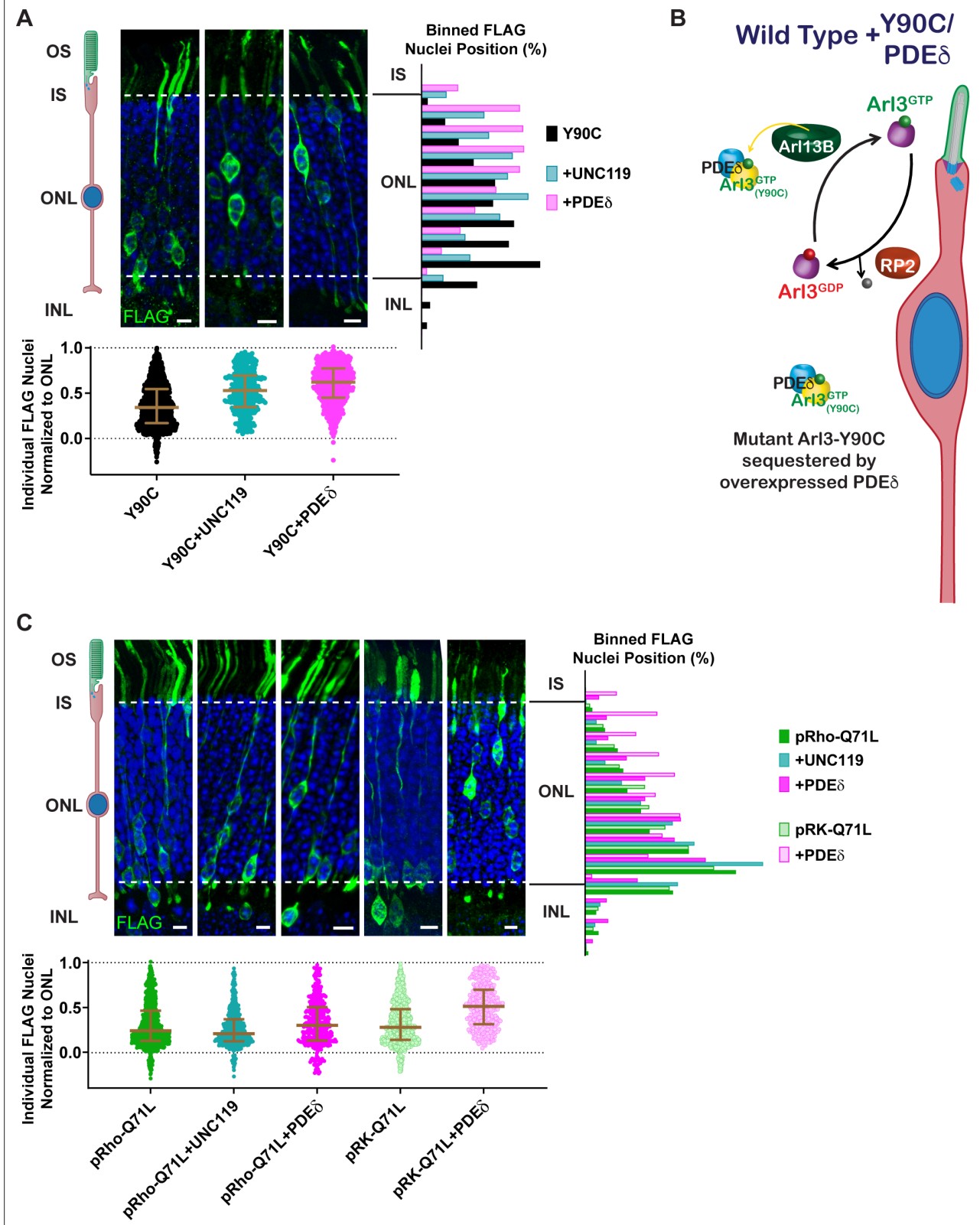

**Figure 6.** Arl3-driven nuclear migration defects are rescued by overexpressing Arl3 effectors in rods. (**A**) Representative retinal cross-sections from Arl3-Y90C-FLAG co-expression with either the Arl3 effectors UNC119a or PDEδ. (**B**) Cartoon model illustrating the impact of Arl3 effector expression in the presence of Arl3-Y90C-FLAG in 'immature' rod photoreceptors. Arl3 effectors reduce active Arl3 levels by sequestering Arl3-GTP. The ciliary Arl3-GTP gradient is restored allowing for normal nuclear positioning. (**C**) Representative retinal cross-sections from high (pRho) or low (pRK) Arl3-Q71L-FLAG co-

*Figure 6 continued on next page*

Figure 6 continued

expressed with either the Arl3 effectors UNC119a or PDEδ. FLAG (green) and Hoechst (blue). Scale bars, 5 µm. Nuclear position of electroporated rods represented as described in *Figure 1*.

The online version of this article includes the following source data and figure supplement(s) for figure 6:

**Source data 1.** Raw western blot images.

**Figure supplement 1.** Co-expression of UNC119a-myc or Arl13B-GFP decreases Arl3-Y90C-FLAG activity in cells.

## Overexpression of lipidated cargos destined for the cilium can rescue the Arl3-Y90C migration defect

We found that chaperone availability is a critical aspect of the migration defect, so we wanted to test whether adding excess lipidated cargo, thereby shifting the equilibrium of chaperone binding from Arl3-GTP to cargo, could also rescue the Arl3-Y90C phenotype. First, we co-expressed two different ciliary lipidated cargos, myc-INPP5E or NPHP3-myc, and found that they both rescued the Arl3-Y90C nuclear migration defect (*Figure 7A*). We also tested non-farneslyated INPP5E-C644A, which has reduced binding to PDEδ (*Figure 7—figure supplement 1*) and found that the Arl3-Y90C migration defect persisted. This lipid dependency confirms that cargo-chaperone binding is critical for rescue. Next, we tested a non-ciliary PDEδ cargo, Rnd1 (*Hanzal-Bayer et al., 2002*), that we confirmed binds PDEδ in a lipid-dependent manner (*Figure 7—figure supplement 1*). Interestingly,

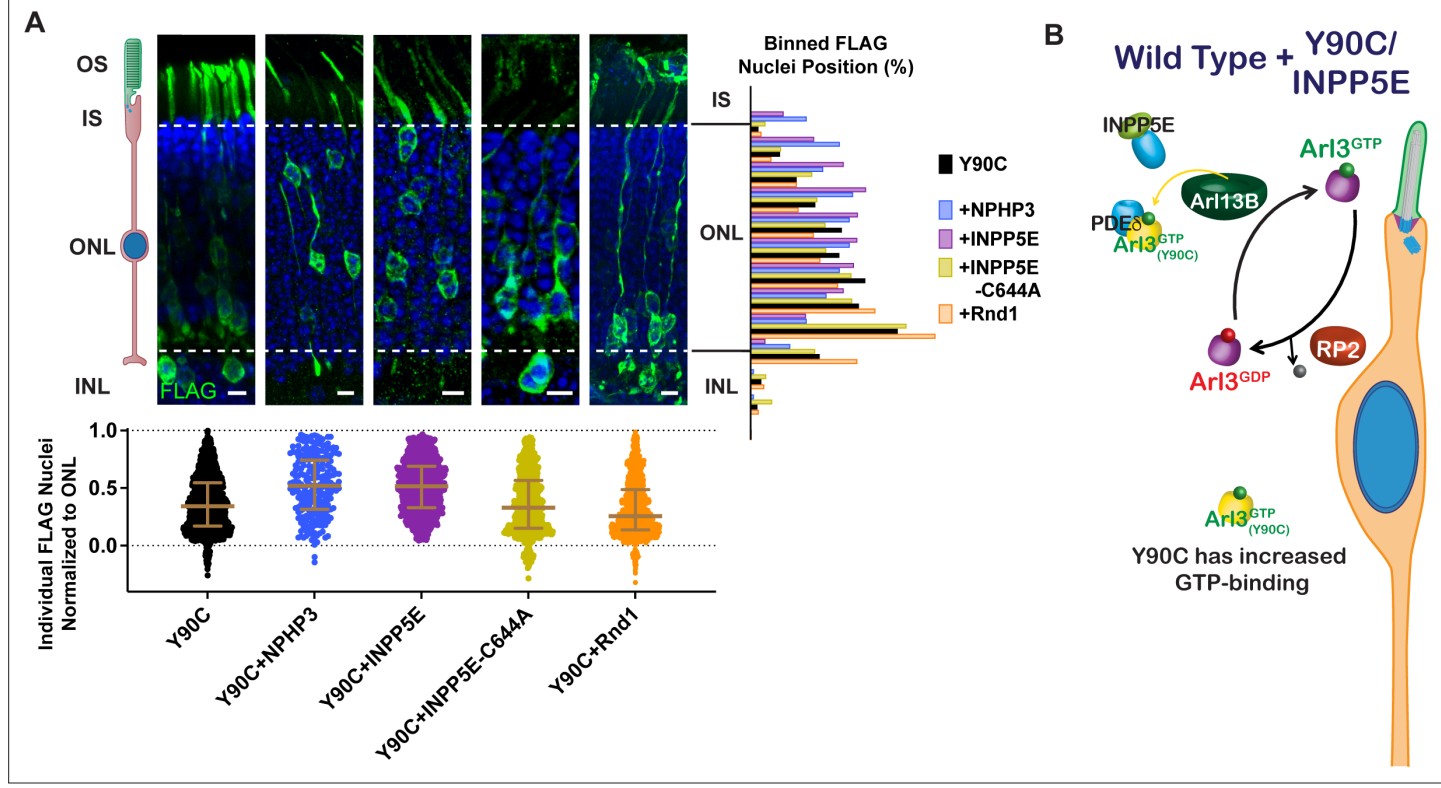

**Figure 7.** Arl3-Y90C migration defect is rescued by overexpressing lipidated cargos destined for the cilium. (**A**) Representative retinal cross-sections from Arl3-Y90C-FLAG co-expression with different lipidated cargos: NPHP3, INPP5E, or non-lipidated INPP5E-C644A (see *Figure 7—figure supplement 1*), and Rnd1 (a PDEδ cargo, see *Figure 7—figure supplement 1*). FLAG (green) and Hoechst (blue). Scale bars, 5 µm. Nuclear position of electroporated rods represented as described in *Figure 1*. (**B**) Cartoon model illustrating the impact of ciliary lipidated cargo expression in the presence of Arl3-Y90C-FLAG in 'immature' rod photoreceptors. Specific delivery cargos to the cilium cause a sufficient local enrichment of Arl3 effector (e.g., PDEδ) for removal of Arl3-Y90C from Arl13B. Even with the presence of aberrant Arl3-Y90C-GTP in the rod cell body (orange) restoration of the endogenous Arl3 (purple) GTPase cycle allows for normal nuclear positioning.

The online version of this article includes the following source data and figure supplement(s) for figure 7:

**Source data 1.** Raw western blot images.

**Figure supplement 1.** Lipid-dependent binding of INPP5E and Rnd1 to PDEδ.

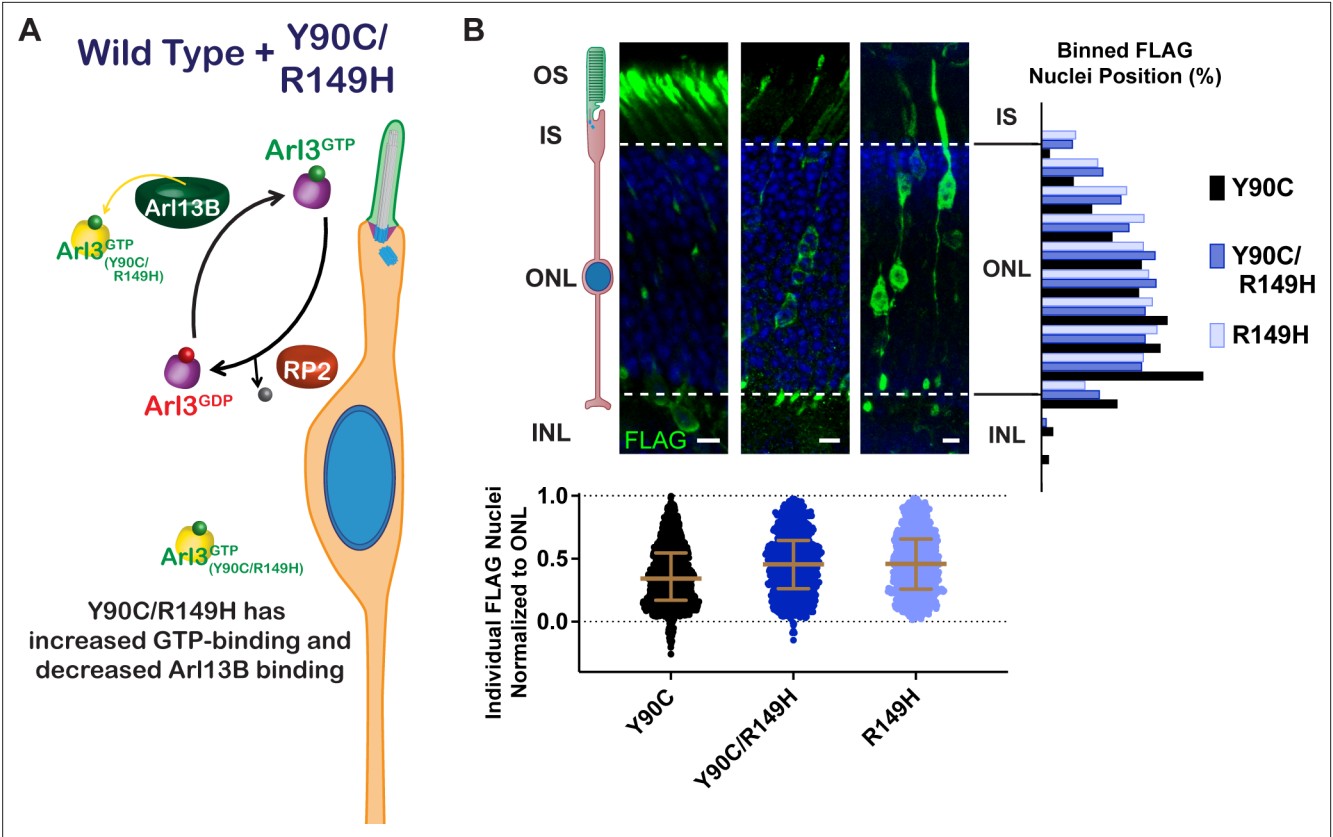

**Figure 8.** Restoring the endogenous Arl3 cycle is sufficient for normal nuclear migration in the presence of aberrant Arl3 activity. (**A**) Cartoon model depicts the Arl3 GTPase cycle in 'immature' rod photoreceptors in the presence of Arl3-Y90C/R149H. Endogenous Arl3 (purple) can restore the Arl3-GTP ciliary gradient (green) even in the presence of aberrant Arl3-Y90C/R149H-GTP in the rod cell body (orange). (**B**) Representative retinal cross-sections from Arl3-Y90C-FLAG-, Arl3-Y90C/R149H-FLAG-, or Arl3-R149H-FLAG-expressing rod photoreceptors. FLAG (green) and Hoechst (blue). Scale bars, 5 μm. Nuclear position of electroporated rods represented as described in *Figure 1*.

we found that co-expression of myc-Rnd1 was unable to rescue the Arl3-Y90C nuclear migration defect (*Figure 7A*). Together, our data show that only ciliary, chaperone-interacting cargo rescues the migration defect, which suggests that local increases in chaperone at the cilium may restore the Arl3 gradient in the cell by directly chelating Arl3-Y90C from Arl13B allowing endogenous Arl3 to activate normally (*Figure 7B*).

## Restoring the endogenous Arl3 cycle is sufficient for normal nuclear migration in the presence of aberrant Arl3 activity

Our results show that a Arl3-Y90C/R149H double mutant maintains aberrant activity (*Figure 4F*) but is unable to bind to Arl13B (*Figure 3A*). This double mutant allows us to directly test the importance of the endogenous Arl3 ciliary gradient in rods since endogenous Arl3 should now be activated in the cilium by Arl13B even though aberrant activation of Arl3-Y90C/R149H will remain present throughout the cell (*Figure 8A*). In contrast to Arl3-Y90C, in vivo expression of Arl3-Y90C/R149H does not cause the migration defect, behaving similar to the Arl3-R149H single mutant (*Figure 8B*). So even though the presence of Arl3-Y90C/R149H has more overall Arl3 activity within rods, the consequences on nuclear migration are blunted due to the Arl3-GTP gradient in the cilium set by endogenous Arl3 interaction with Arl13B. This result underscores the importance of the ciliary gradient of active Arl3 for normal neuronal migration.

## Discussion

### Arl3 GTPase function

A recent flurry of papers identified multiple Arl3 variants that cause either dominant or recessive retinal disease (*Alkanderi et al., 2018*; *Fu et al., 2021*; *Holtan et al., 2019*; *Ratnapriya et al., 2021*; *Sheikh et al., 2019*; *Strom et al., 2016*). These studies highlighted the necessity of Arl3 for photoreceptor health and raised the possibly that different biochemical alterations in Arl3 might produce divergent pathobiological underpinnings. In this study, we focused on the biochemical and cellular consequences of dominant mutations in Arl3 and found that aberrant activity in rods causes a developmental nuclear migration defect. The two dominant Arl3 variants, D67V and Y90C, phenocopied the constitutively active Q71L, previously shown to result in mislocalization of rod nuclei into the INL (*Wright et al., 2016*). However, these dominant mutations had very different GTPase behaviors: the D67V behaved as a constitutively active GTPase and the Y90C functioned as a fast cycling GTPase.

Previous in silico analyses predicted that the D67V mutation would decrease affinity to the Arl3-binding partners RP2, UNC119a, and Arl13B (*Ratnapriya et al., 2021*). In fact, we found that Arl3-D67V forms a stable complex with RP2, similar to the constitutively active Q71L variant (*Figure 4C*); but appears to subtly change the binding properties toward the effectors PDEδ and UNC119. Even though Arl3-D67V can bind PDEδ and RP2, it does not bind them both at the same time, a Arl3-Q71L feature shown both here and previously (*Figure 2—figure supplement 2*; *Veltel et al., 2008b*). We also found that Arl3-D67V disrupted UNC119 binding (*Figure 4—figure supplement 1*). This effector specificity is surprising considering how similar the binding interfaces are in the UNC119a-Arl2/3 and PDEδ-Arl2 crystal structures (*Hanzal-Bayer et al., 2002*; *Ismail et al., 2012*). Further study of the Arl3-D67V variant could provide insight into both the structural and functional differences between PDEδ and UNC119.

Tyrosine 90 is located right in the center of the G-domain, and this site is not strictly conserved throughout the small GTPase family. Previous predictions were that Y90C would result in haploinsufficiency due to destabilized binding to effectors or protein instability (*Holtan et al., 2019*; *Strom et al., 2016*). Instead, we discover that the Arl3-Y90C variant is behaving like a fast cycling GTPase as it has GEF-independent GTP binding (*Figure 2*). Our results highlight that fast cycling mutations are also dominant negative—normal activation of the endogenous GTPase is altered by Arl3-Y90C binding to Arl13B in the cilia. Interestingly, while the dominant negative feature of Arl3-Y90C was necessary it was not sufficient to produce a migration defect suggesting that aberrant activity outside the cilia plays a role.

We developed an easy and useful in vivo crosslinking assay to determine nucleotide-binding state of small GTPase proteins within cells. This technique allows for a snapshot of activity that cannot be perturbed during lysis and pulldown and provides quantitative information about effector-binding ratios. We showed that Arl3-Y90C has aberrant activity in the cells, although not as robust as Q71L, and is able to hydrolyze GTP as it does not make a stable complex with RP2 (*Figure 4*). Despite the higher levels of Arl3-Y90C activity in the cell, we actually find that reactivating endogenous Arl3 in the presence of this activity prevents the nuclear migration defect in rods. This suggests that the defect is not caused by an increase in overall Arl3 activity, but a disruption of proper Arl3-GTP localization within the cell. We found only co-expression of ciliary lipidated cargos could restore the migration defect as they deliver Arl3 effectors directly to the cilia reactivating endogenous Arl3 at that site (*Figure 7*).

We demonstrate that more active Arl3 must be present in the cilium than the cell body for proper migration of rod nuclei during retinal development. This requirement for a ciliary Arl3-GTP gradient seems very similar to the Ran-GTP gradient important for nuclear transport (*Sorokin et al., 2007*). Movement down this gradient acts as a driving force behind cargo delivery into the cilium, so altering rather than removing the Arl3 gradient would have a distinct impact on cargo behavior. When Arl3 activity levels are increased in the cell body relative to the cilium, the driving force reverses, and cargo is no longer delivered. It is possible that disrupting delivery results in downregulation of the cargo. This might explain why absence of Arl3 activity does not cause a migration defect (*Figure 5*), as there would be no driving force on the cargo and its widespread delivery could result in sufficient accumulation at the cilium.

## Neuronal migration

We found that mutations in Arl3 can cause a mislocalization of rod photoreceptor nuclei toward the basal ONL and into the INL. Our results show a satisfying alignment between biochemical behavior, genetic presentation, and migration phenotype; however, our studies did not address whether the migration defect caused by the aberrant Arl3 activity is sufficient to cause degeneration. The knockout of RP2, resulting in increased Arl3 activity, causes progressive photoreceptor degeneration and has a population of rods mislocalized to the INL (*Li et al., 2013*). However, RP2 gene therapy in adult mice slowed degeneration, suggesting that the rod nuclear mislocalization may not be pathogenic (*Li et al., 2013*; *Mookherjee et al., 2015*). Nevertheless, it was previously shown that basal displacement of rod nuclei coincides with retinal disfunction including defects in synapse formation (*Aghaizu et al., 2021*; *Yu et al., 2011*), so even if the rod migration defect does not directly cause cell autonomous degeneration it could affect the physiological function of rods and therefore retinal health. The exact pathobiological outcomes of displaced photoreceptor nuclei remain to be fully understood.

A related question is why the two dominant mutations cause retinal degeneration but not ciliopathies. Arl3 is expressed in all ciliated cells, and some recessive Arl3 mutations do cause syndromic disease. In fact, mutations in Arl13B, the Arl3 GEF, cause the syndromic ciliopathy Joubert syndrome (*Cantagrel et al., 2008*) and can cause a reduction of Arl3 activation (*Gotthardt et al., 2015*). The Arl3 GAP RP2, on the other hand, causes non-syndromic retinal degeneration (*Breuer et al., 2002*), providing more evidence that the impact of excess active Arl3 on cells is more detrimental for retinal photoreceptors than other ciliated cells throughout the body. Further studies are needed to determine if the migration phenotype is present in other neurons or if the mechanism linking Arl3 function to nuclear translocation is photoreceptor specific.

Finally, while our data demonstrate that the cilium is involved in apical nuclear translocation of rod photoreceptors, the precise ciliary signal remains unknown. Arl3 is known to traffic many lipidated signaling proteins to the ciliary outer segment in mature mouse rods (*Hanke-Gogokhia et al., 2016*; *Wright et al., 2016*). Most of these candidates are involved in mediating the light response and not known to be involved in nuclear migration, which is a developmental process that ends at P10 for mouse rod photoreceptors (*Aghaizu et al., 2021*). 'Immature' migrating rods are ciliated, but the developmental function as well as whether any Arl3-dependent ciliary cargos are enriched within the cilia remains unknown. In the brain, primary cilia have previously been shown to play a role in neuronal nuclear migration, with studies implicating both Arl13B (*Higginbotham et al., 2012*) and cAMP signaling (*Stoufflet et al., 2020*). Our study now links Arl3 to nuclear migration and the next step will be to identify the lipidated cargo relying on the ciliary Arl3 gradient within photoreceptors during development.

# Materials and methods

## Animals

Mice were handled following protocols approved by the Institutional Animal Care and Use Committees at the University of Michigan (registry number A3114-01). Albino CD-1 wild-type mice used in electroporation experiments were ordered from Charles River Laboratories (RRID:IMSR_CRL:022; Mattawan, MI). All mice were housed in a 12/12-hr light/dark cycle with free access to food and water.

## DNA constructs

For rod-specific expression, sequences were cloned between 5′ *AgeI* and 3′ *NotI* cloning sites into a vector with of 2.2 kb of the mouse rhodopsin promoter, which was originally cloned from pRho-DsRed, a gift from Connie Cepko (Addgene plasmid #11156; RRID:Addgene_11156). FLAG-tagged Arl3 was created by cloning the entire human *ARL3* sequence from pDEST47-ARL3-GFP, a gift from Richard Kahn (Addgene plasmid #67397; RRID:Addgene_67397) followed by a AVPVDSRGSRA linker and a C-terminal 3X FLAG sequence (Sigma-Aldrich). All mutations in Arl3 were created using PCR mutagenesis (*Weiner et al., 1994*). The Arl3-Q71L-FLAG was also cloned into a plasmid with the human rhodopsin kinase promoter (originally cloned from hRK:GFP AAV, a gift from Tiansen Li). For chaperone overexpression: either human *PDE6D* (NM_002601.4; Dharmacon MHS6278-202829730) or human *UNC119A* (NM_005148; Origene RC203758) was cloned in front of a T2A site followed by mCherry to allow for co-translational cleavage and expression. For cargo overexpression: N-terminal

myc tags were placed in front of mouse *INPP5E* (*Fansa et al., 2016*) and human *RND1* cloned from GFP-Rnd1, a gift from Channing Der (Addgene plasmid #23227), and C-terminal myc tag was placed in front of human *NPHP3* (GeneCopoeia GC-H2370). All mutations to replace the lipidated cysteine with alanine were made using PCR mutagenesis (*Weiner et al., 1994*).

Every protein expressed in mouse rods was also cloned into pEGFP-N1 (Clontech) for expression in AD-293 cells using the *AgeI* and *NotI* cloning sites within the vector to replace EGFP with the tagged proteins, except for UNC119A which was tagged with C-terminal myc rather than in front of a T2A site for cell culture expression. Additionally, *Arl13B* was a cloned from mouse cDNA into pcDNA3-EGFP plasmid, a gift from Doug Golenbock (Addgene plasmid #13031; RRID: Addgene_13031) and pEGFPN1-HA-RP2 was cloned from mouse *Rp2* cDNA (Dharmacon MMM1013-202842815).

GST-PDEδ was cloned by adding human PDEδ (NM_002601.4; Dharmacon MHS6278-202829730) to an expression vector with N-terminal GST and 6XHis: pET28GST-LIC was a gift from Cheryl Arrowsmith (Addgene plasmid #26101; RRID: Addgene_26101). A complete list of primers can be found in *Supplementary file 1*.

## In vivo electroporation of mouse retinas

DNA constructs were electroporated into the retinas of neonatal mice using the protocol as originally described by *Matsuda and Cepko, 2004*, but with modifications as detailed by *Pearring et al., 2015*. Briefly, P0–P2 mice were anesthetized on ice, had their eyelid and sclera punctured at the periphery of the eye with a 30-gauge needle, and were injected with 0.25–0.5 µl of 2 µg/µl concentrated plasmid with 0.5 µg/µl of pRho-mCherry (*Pearring et al., 2014*) or pRho-eGFP-Cb5$_{TM}$ (*Baker et al., 2008*) as an electroporation marker in the subretinal space using a blunt-end 32-gauge Hamilton syringe. The positive side of a tweezer-type electrode (BTX) was placed over the injected eye and five 100 V pulses of 50 ms were applied using an ECM830 square pulse generator (BTX). Neonates were returned to their mother until collection at P21.

## Immunofluorescence

Eyes were enucleated at P21 after sacrifice by $CO_2$ inhalation and fixed in 4% paraformaldehyde in phosphate-buffered saline (PBS) at room temperature for 1 hr before washing with PBS. Eye cups with patches of mCherry expression were embedded in 4% agarose (Thermo Fisher Scientific, BP160-500) and cut into 100-µm thick sagittal sections. After blocking with 5% donkey serum (Thermo Fisher Scientific NC0629457) and 0.2% Triton X-100 in PBS for 1 hr at room temperature, free floating sections were incubated with mouse anti-FLAG clone M2 antibodies (1:2000 in the donkey serum blocking solution, Sigma-Aldrich Cat#F1804, RRID:AB_262044), overnight at 4°C, washed with PBS, and then incubated for 1 hr at room temperature with donkey secondary antibodies conjugated to 488 or 647 (Jackson ImmunoResearch Labs Cat# 715-545-151, RRID:AB_2341099) and 10 µg/ml Hoechst 33342 (Fisher Scientific BDB561908) in the donkey blocking solution before washing and mounting on slides with Immu-Mount (Thermo Fisher Scientific) and 1.5-mm coverslips (EMS).

## Image analysis

Images were acquired using a Zeiss Observer 7 inverted microscope equipped with a ×63 oil-immersion objective (1.40 NA), LSM 800 confocal scanhead outfitted with an Airyscan superresolution detector controlled by Zen 5.0 software (Carl Zeiss Microscopy). Manipulation of images was limited to adjusting the brightness level, image size, rotation, and cropping using FIJI (ImageJ, https://imagej.net/Fiji). Rod nucleus locations were measured using Imaris software (Bitplane). The 3D planes defining the bottom and top of the nuclear stack as well as the location of each nucleus of a FLAG-positive cell were chosen by hand and recorded. MATLAB (Mathworks) code was written to convert each nuclear location within the z-stack to its location between the two planes and then assigned a normalized value corresponding to the shortest distance to the plane defining the bottom of the nuclear stack. All phenotypes were measured using images taken from at least three independent retinas.

## Cell culture

All cell culture experiments were performed using AD-293 cells (Agilent Technologies, 240085; RRID:CVCL_9804) maintained at 37°C in 5% $CO_2$. The AD-293 cells were authenticated by the commercial source by morphology, trypan-blue dye exclusion, and viable cell count, and provided

free of microbial contamination as determined by sterility culture testing in M-TGE and YM broth, and mycoplasma testing by PCR. Transient transfections were performed 1 day after seeding $1 \times 10^6$ cells in a 10-cm plate and experimental collection was performed 2 days after transfection. 5 µg of plasmid DNA was transfected per plate by incubating the DNA with 10 µg polyethyleneimine (Sigma, 408727) in serum-free media for 10 min before being added dropwise to cells that had been changed to serum-free media 1 hr prior.

## Pulldowns

GST-PDEδ and GST alone were each expressed and purified from BL21(DE3) *E. coli* (NEB) after adding 0.25 mM isopropyl β-d-1-thiogalactopyranoside (IPTG) for 4 hr at 28°C, lysing the cells (50 mM Tris pH 8, 50 mM NaCl, 5 mM $MgCl_2$, 5 mM β-mercaptoethanol (BME), and cOmplete EDTA-free protease inhibitor [Millipore Sigma, 1183617001]), and eluting from a column with Ni-NTA beads (Thermo Fisher, 88221) using 100 mM imidazole pH 8. GST-PDEδ (50 µg) was then crosslinked to 50 µl of magnetic glutathione beads (Sigma-Aldrich, G0924) using disuccinimidyl suberate (Thermo Scientific, 21555) and the beads were used to pulldown active Arl3 from AD-293 cell lysates 48 hr after transient transfection with Arl3-FLAG mutants. Briefly, cells were lysed (20 mM Tris pH 7.5, 200 mM NaCl, 5 mM $MgCl_2$, 0.5% Igepal [Fisher Scientific, AAJ19628AP], and cOmplete protease inhibitor cocktail), cleared by centrifugation at 14,000 × *g* for 10 min, incubated with GST beads for 10 min at 4°C, incubated with GST-PDEδ beads for 45 min at 4°C, washed with Buffer M (20 mM Tris pH 7.5, 5 mM $MgCl_2$, 100 mM NaCl, 3 mM BME, 1% glycerol), and then heated to 55°C for 10 min in Laemmli sample buffer. When testing for the effect of $Mg^{2+}$ and/or excess GTP on the amount of active Arl3, final concentrations of 10 mM EDTA in the presence or absence of 10 mM GTP (Thermo Fisher, R0461) or 65 mM $MgCl_2$ in the presence of 10 mM GTP were added to the lysate and vortexed before shaking at 30°C for 15 min. To stop the GTP loading in the presence of 10 mM EDTA, $MgCl_2$ to a final concentration of 65 mM was added before the pulldown.

## In vivo crosslinking and FLAG immunoprecipitations

10-cm confluent plates of transiently transfected AD-293 cells were washed well with PBS and then incubated in 1 mM disuccinimidyl suberate (Thermo Scientific, 21555) in PBS at 37°C for 10 min. The reaction was quenched by adding Tris Buffer pH 8, to a final concentration of 100 mM for 15 min at room temperature. After washing again with PBS, cells were collected and lysed in 50 mM Tris pH 7.5, 100 mM NaCl, 5 mM $MgCl_2$, 0.5% Igepal, 2% glycerol with cOmplete protease inhibitor cocktail. Immunoprecipitations were performed by adding 15 µl of magnetic FLAG beads (Sigma-Aldrich, M8823) to the lysates and rotating at 4°C for 2 hr before washing three times with a high salt buffer (50 mM Tris pH 7.5, 500 mM NaCl, 5 mM $MgCl_2$, 2% glycerol) and then eluting by shaking with 15 µl of 100 µg/µl of 3× FLAG peptide (Sigma-Aldrich, F4799) in the original lysis buffer for 30 min at 4°C. A minimum of three replicates was performed for each Arl3-FLAG construct and amount of Arl13B-GFP binding to each Arl3-FLAG is shown in *Figure 3—figure supplement 3*.

## Western blotting

Sodium dodecyl sulfate–polyacrylamide gel electrophoresis (SDS–PAGE) using AnykD Criterion TGX Precast Midi Protein Gels (Bio-Rad) was followed by transfer at 90 mV for 90 min onto Immun-Blot Low Fluorescence PVDF Membrane (Bio-Rad). Membranes were blocked using Intercept Blocking Buffer (LI-COR Biosciences). The antibodies used for western blotting were mouse anti-FLAG clone M2 (1:1000, Sigma-Aldrich Cat#F1804, RRID:AB_262044), polyclonal rabbit anti-Arl3 (1:1000, Novus Cat# NBP1-88839, RRID:AB_11028976), monoclonal anti-GFP JL-8 (1:1000, Takara Bio Cat# 632380, RRID:AB_10013427), polyclonal rabbit anti-RP2 (1:1000, Sigma-Aldrich Cat# HPA000234, RRID:AB_1079831), and monoclonal anti-MYC clone 9B11 (1:1000, Cell Signaling Technology Cat# 2276, RRID:AB_331783). Primary antibodies were diluted in 50%/50% of Intercept/PBS with 0.1% Tween-20 (PBST) and incubated overnight rotating at 4°C. The next day, membranes were rinsed three times with PBST before incubating in the corresponding secondary donkey antibodies conjugated with Alexa Fluor 680 or 800 (LiCor Bioscience) in 50%/50%/0.02% of Intercept/PBST/SDS for 2 hr at 4°C. Bands were visualized and quantified using the Odyssey CLx infrared imaging system (LiCor Bioscience). Images of the uncropped western blots can be found in Source Files.

## Mass spectroscopy

Samples analyzed by mass spectroscopy were produced using the above protocols for transient transfection, in vivo crosslinking, and FLAG immunoprecipitation. After elution from the beads with 1× Laemmli sample buffer and brief gel electrophoresis, ~1 × 1 mm gel pieces were subjected to reduction (20 mM dithiothreitol (DTT), 50 mM ammonium bicarbonate for 60°C for 15 min), alkylation (50 mM iodoacetamide SigmaUltra I1149 in 50 mM ammonium bicarbonate in the dark at RT for 1 hr), in-gel tryptic digestion (250 ng Trypsin-Lys-C mix Promega V5072 in 50 mM ammonium bicarbonate overnight at 37°C), and peptide extraction (0.2% trifluoroacetic acid in 50% acetonitrile at RT for 30 min). After drying by SpeedVac, the peptides were analyzed with a nanoAcquity UPLC system (Waters) coupled to an Orbitrap Q Exactive HF mass spectrometer (Thermo Fisher Scientific) employing the LC–MS/MS protocol in a data-independent acquisition mode. The peptides were separated on a 75 µm × 150 mm, 1.7 µm C18 BEH column (Waters) using a 90-min gradient of 8–32% of acetonitrile in 0.1% formic acid at a flow rate of 0.3 ml/min at 45°C. Eluting peptides were sprayed into the ion source of the Orbitrap Q Exactive HF at a voltage of 2.0 kV. Progenesis QI Proteomics software (Waters) was used to assign peptides to the features and generate searchable files, which were submitted to Mascot (version 2.5) for peptide identification and searched against the UniProt reviewed mouse database (September 2019 release, 17008 entrees). Only proteins with two or more peptides and protein confidence $p < 0.05$ were assigned as confidently identified.

## Statistical analysis

To analyze the location of Arl3 mutant expressing rod nuclei, the skewness of the nuclei in each individual z-stack was calculated using GraphPad Prism. The pooled skewness values for each electroporation condition were then compared using a one-way analysis of variance (ANOVA) and a significant difference ($p < 0.0001$) was found among the means. Dunnett's multiple comparisons tests were then done to determine whether each condition had a significantly different nuclear localization skew than wild-type Arl3-FLAG-expressing rods. The skewness and p value of the multiple comparison test for each condition are listed in *Table 1*.

To analyze the Arl13B immunoprecipitation experiments performed in AD-293 cells expressing Arl3-FLAG mutants (*Figure 4E*), Arl13B-GFP bound was normalized to Arl3-FLAG in the same lane. The bands analyzed were a part of eight separate experiments; every experiment included one Y90C lane. If no Arl13B-GFP is bound it results in a 0 value, so a Wilcoxon signed-rank test was performed to see whether the data for amount of Arl13B bound to each Arl3-FLAG mutant were significantly different from 0. Arl3-Y90C-FLAG resulted in \*\*$p = 0.0078$.

For analysis of the amount of active Arl3 complexes formed in AD-293 cells expressing Arl3-FLAG mutants (*Figure 4E*), the ~55 and ~45 kDa bands within multiple biological replicates were first normalized to the intensity of their own uncrosslinked Arl3-FLAG band and then normalized to the Arl3-Q71L-FLAG complex bands on the same gel. The bands analyzed were a part of seven separate experiments; every experiment included one T31N lane and one Q71L lane, but some included multiple lanes of other mutants. One-way ANOVA between D129N, Y90C, WT, and T31N showed a significant difference among the means ($p < 0.0001$), and Tukey's multiple comparisons were performed.

## Acknowledgements

We are grateful to Mickey Kosloff (University of Haifa, Israel) for excellent feedback on the manuscript. This work was supported by a NIH P30 grant EY007003 (University of Michigan), NIH T32 grant EY013934 (AMT), Matilda E Ziegler Research Award (JNP), Career Development Award (JNP), an Unrestricted Grant (University of Michigan) from Research to Prevent Blindness.

## Additional information

### Funding

| Funder | Grant reference number | Author |
|---|---|---|
| Research to Prevent Blindness | Career Development Award | Jillian N Pearring |
| E. Matilda Ziegler Foundation for the Blind | Research Award | Jillian N Pearring |
| National Eye Institute | T32 Postdoctoral Award EY013934 | Amanda M Travis |

The funders had no role in study design, data collection, and interpretation, or the decision to submit the work for publication.

### Author contributions

Amanda M Travis, Conceptualization, Formal analysis, Funding acquisition, Investigation, Methodology, Writing – original draft, Writing – review and editing; Samiya Manocha, Jason R Willer, Investigation, Writing – review and editing; Timothy S Wessler, Software; Nikolai P Skiba, Formal analysis, Investigation, Writing – review and editing; Jillian N Pearring, Conceptualization, Supervision, Funding acquisition, Methodology, Writing – original draft, Writing – review and editing

### Author ORCIDs

Jillian N Pearring http://orcid.org/0000-0002-5352-2852

### Ethics

This study was performed at the University of Michigan, following strict accordance with the recommendations in the Guide for the Care and Use of Laboratory Animals and accreditation from the Association for Assessment and Accreditation of Laboratory Animal Care (AAALAC) International. Mice were handled following protocols approved by the Institutional Animal Care and Use Committees at the University of Michigan (registry number A3114-01). All mice were housed in a 12/12-hr light/dark cycle with free access to food and water and every effort was made to minimize suffering.

### Decision letter and Author response

Decision letter https://doi.org/10.7554/eLife.80533.sa1
Author response https://doi.org/10.7554/eLife.80533.sa2

## Additional files

### Supplementary files

• Supplementary file 1. Primer sequences and usage.
• MDAR checklist

### Data availability

All data generated or analyzed for this study are included in the manuscript and supporting files; source data files have been provided for all figures. The MATLAB code used for normalization of nuclear position is available on GitHub (*Travis et al., 2022*; copy archived at swh:1:rev:72399998d5574799cc7be095463400c1a3363e36).

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
