## [Editor Report]

This paper will be of interest to scientists interested in studying ciliogenesis and specifically vertebrate photoreceptors, which are specialized cilia. The study shows that mutations in the small GTP binding protein ARL3 known to cause dominant inherited human retinal dystrophies result in ARL3 hyperactivity, disrupt the normal ciliary gradient of ARL3 activity, and alter nuclear migration affecting retinal development.

---

## [Decision Letter]

**Decision letter after peer review:**

Thank you for submitting your article "Disrupting the ciliary gradient of active Arl3 affects rod photoreceptor nuclear migration" for consideration by *eLife*. Your article has been reviewed by 3 peer reviewers, and the evaluation has been overseen by a Reviewing Editor and Piali Sengupta as the Senior Editor.

Essential revisions:

As you will see that all the reviewers agree that the paper is of interest to cell biologists studying ciliogenesis and specifically vertebrate photoreceptors. Overall, they mention that the experiments are properly controlled, executed, and analyzed. The development of a method to analyze snapshots of the interaction between ARL3 and its interactors is also a strength of the paper. However, significant concerns have been raised by all the reviewers. We highlight two major concerns at the heart of the paper's current message that need to be addressed before publication.

1. How does nuclear migration failure impact ciliogenesis in the outer segment? The reviewers suggest that further analysis of the trafficking defects of effectors and cilium length for electroporated variants could better link cilium/trafficking defects with the nuclei migration phenotypes.

2. Testing alternative hypotheses for the Y90C mutant with regards to (a) sequestration of the GEF ARL13B in preventing activation of the endogenous ARL3, and (b) effects on subcellular localization. The reviewers suggest that these alternative models might be tested by the rescue of ARL3 Y90C driven phenotype upon co-expression of ARL13B and detailed imaging and analysis for the ARL3 variants/potential effectors in photoreceptor subcellular segments.

Please also provide a point-by-point response to the remaining critiques of the reviewers.

*Reviewer #1 (Recommendations for the authors):*

1) Figure 1: It would be better to have a description of what the value means on the left side as well as the bottom side of the scatter plot. The same is true for the histogram. For the histogram, the labels that indicate which mutant the authors expressed should go to the right side of the histogram, rather than the left side, as the labels are at the place where the X-axis of the histogram is located.

Also, what is the unit of the y-axis of the histogram? Is this indicating the number? Or percentage?

2) Figure 2D: The figure legend indicated that the scale bar is 5µm, but there is no scale bar in the image.

3) Figure 3E: Why there are so many dots here? Are these mixture of the normalized signal intensity of the bands corresponding to UNC119 and BART? If this comes from multiple experiments, how many experiments did the author perform? Are the experiments technical replicates? Or biological replicates? Why does the number of dots differ among the samples?

In Figure 3D, the signal intensity of the 45kDa and 55kDa in D129N cells does not look as strong as the bands in Y90C cells. However, the quantification shown in Figure 3E indicated that the signal intensity of the bands in D129N cells is much stronger than those of Y90C cells. Is this because the authors normalized the intensity by the intensity of the FLAG-band?

4) Figure 3F, Figure 3G, and Figure 4A: It would be great if the authors could have quantification just like the authors did in Figure 3E.

5) Line59: R991 should be R99I.

6) Line 189: "It was unable to form complexes with expressed with ARL13B-GFP".

Should this be "It was unable to form complexes when expressed with ARL13B-GFP"?

7) Line 283-284: "but appears to subtly change the binding properties toward the effectors PDEδ and UNC119." What is this sentence referring to? Figure 2A? or Figure 3C? The affinity towards PDEδ seems to be similar between Q71L and D67V in Figure 2A?

8) Line 353: I might have missed it, but I did not find what expression vectors did authors use for the injection to the eye. Is the backbone of the expression vector pRho-DsRed? (as indicated in line 356)? Or authors just took the promoter out of the pRho-DsRed? In this case, what is the backbone of the expression vector?

9) Line 440: Where is the GTP coming from? Is the catalog number lacking? Also, many other chemicals lack the catalog number. It would be better to include those.

10) Line 439-441: Wording is a little confusing. Maybe "10mM EDTA in the presence or the absence of 10mM GTP, or 65mM MgCl2 in the presence of 10mM GTP"?

11) Line 465: PBST. Is this PBS containing 0.1% Tween-20?

12) 477: 37oC. the unit (degree Celsius) is not displayed correctly.

13) Line 497-501: It is not very clear how the authors measured the intensity. Did the author use Li-COR software? Or Image J? Did the author subtract the background?

*Reviewer #2 (Recommendations for the authors):*

1. The authors never examine actual ciliogenesis in their retinal model – nuclear migration is the only readout. First, some description of how nuclear migration impacts retinal development should be included in the introduction. More importantly, however, the authors should specifically show the morphology of the cilia in cells expressing the various Arl3 mutants. This is particularly important for experiments in which the expression of effectors or cargo relieves the nuclear migration phenotype induced by the mutants. Do effectors and/or cargo reach the cilia, and are they distributed normally, even if nuclear migration is rescued? Immunostaining for PDEd, UNC199A, INPP5E, and NPHP3 would be essential for this purpose.

2. The authors should expand their discussion of the Y90C mutant to include its action as a dominant-negative, and the complex nature of its features. It is clearly not simply a rapid cycling mutant.

3. Do any of these mutants have phenotypes in non-retinal ciliated cells? If this is known it could simply be described, but if not they could be tested simply in cell culture models.

*Reviewer #3 (Recommendations for the authors):*

– The word "gradient" is stated throughout the text, including in the title. However, none of the experiments ever directly assess this. And is it really a gradient? Or just subcellular compartmentalization segregating the specialized cilium of photoreceptors and the cell body?

– Since spatial localization of Arl3, Arl13B, RP2, and Arl3 effectors PDEd and UNC119, seem to be key in the overall model of differential subcellular activation of Arl3, it should be better highlighted in the introduction where these proteins are localized in photoreceptors specifically, with the corresponding original references.

– Line 69 (introduction): it is stated that "rod nuclear displacement has also been observed in RP2 KO mouse" with direct references to "Li et al., 2013 and Mookherjee et al., 2015". However, rod nuclei displacement was never directly assessed on these papers. Indeed, there seems to be a phenotype in ONL/OPL/INL of RP2 KO mice, which could potentially be displacement of rod nuclei. This is interesting and should be actually evaluated. I suggest an experiment to directly assess this in this mice KO line or by knocking-down RP2 using in a similar strategy to the one the authors use to deliver transgenes to P0 mice retinas by means of electroporation.

– Line 75 (intro): "using in vivo mouse models…". The word "models" should be substituted by "experiments".

– Line 76 (intro): "show that this aberrant activity disrupts the Arl3-GTP gradient in the cilium". This is actually never directly shown in the manuscript. Combination of multiple experiments together with differential subcellular protein localization, suggest, and suggest only, that this is the case. The wording should be adjusted.

– Line 77 (intro): the rescue experiments were only performed for the Y90C variant, not for the D67V. This should be clear during the text. The way it's written implies that rescue experiments were done for both of the mutations.

– Line 84 (intro): "…in rods using…" modify to "…in mice rods using…"

– More detailed imaging and analysis should be performed for the specialized cilium of photoreceptors (inner segment, connecting cilium and outer segment). Specifically show in more detail the FLAG-staining for each variant. Even though it is stated in the text that "while both Arl3-D67V-FLAG and Arl3-Y90C-FLAG expression and localization were indistinguishable from wild-type Arl3-FLAG", some images do not represent this. For instance, Y90C-FLAG signal seems to be mostly staining outer segments, while D67V seems to mostly stain inner segments, at least in the representative images shown. Is this always the case?

– Analysis should be done to measure cilium length for each condition and also between INL-displaced, basally-displaced and non-displaced rods electroporated with the different variants. In addition, it should be evaluated if there is a correlation between cilium length and rod displacement. This is important in linking both cilium and nuclei, the main point of the manuscript.

– "skew" values are a valuable and sophisticated quantification measure of the phenotype and should be added to Figure 1 (and to all other figures, when applicable) to highlight the phenotype.

– In addition to the skewness analysis, the number of displaced rods normalized to total number of electroporated ones should also be quantified for each condition to better compare and evaluate the phenotype (this is somewhat already shown by the histogram, but adding a percent count will solidify the phenotype finding). This should be done for Figure 1 and all other figures, when applicable. In addition, these percentages should be discussed in the paper. A related question that should be discussed is, since these are dominant variants why isn't the phenotype more dramatic?

– Variant D67V is not as well characterized as Y90C, even though it is introduced with the same significance as the Y90C. What's the model for how D67V is leading to rod displacement, since it does not bind RP2 like the Q71L? Can you add a schematic for this variant to Figure 3H.

– According to your model, Q71L variant is responsible for rod nuclei displacement due to constant binding to RP2, sequestering it and preventing it to inactivate endogenous Arl3 at the base of the cilium. A straight-forward experiment to test this model is to perform a rescue experiment by co-expression of the variant together with RP2, similar to what was done with co-overexpression of UNC119 or PDE δ. In addition, the Q71L [noRP2] triple mutant should also rescue the rod nuclei phenotype, according to your model.

– Perform another rescue experiment of Arl3Y90C driven phenotype by co-overexpression of Arl13B. Increasing the levels of Arl13B should also rescue the rod displacement if there is enough Arl13B to activate endogenous Arl3, according to your proposed model.

– To meet the human relevance justifying your studies (that is, Arl3 variants associated with dominant inherited retinal degeneration), assess potential degenerative phenotype in older mice, such as 2-months old (when severe retinal degeneration is observed in other mouse models, such as Arl3-Q71L transgenic mouse). This experiment would also address a simple potential delay in rod migration, although very unlikely.

– Discuss why the Q71L variant has not been associated with human disease, either retina-specific or more generic ciliopathies.

– Figure 3C: there is an extra band , highlighted by a red arrowhead. This band is mentioned in the figure legend but never in the Results section. Please comment on it.

– Figure 4B: in the rescue experiment of Arl3 T31N/Y90C: do you see a degeneration phenotype if you look at older mice, similar to Arl3KO?

– Consider reorganizing your table by listing the variants by the order they appear in the main text. Consider adding an extra column for "binding effectors: PDEd and UNC119" and state if they bind the specific variant. This table is a great reference.

– Comment should be made as to inconsistencies regarding using human or mouse sequences (for instance, human PDEd, but mouse UNC119). An explanation should be given, such as high level of protein conservation.

– State the ratio of each plasmid when co-expressing multiple proteins.

Figure 5: the beginning of the figure legend "Arl3 driven nuclear…" should be "Arl3-Y90C driven nuclear…".

Figure 5 SpFigure 1: describe red asterisks.

[Editors' note: further revisions were suggested prior to acceptance, as described below.]

Thank you for resubmitting your work entitled "Disrupting the ciliary gradient of active Arl3 affects rod photoreceptor nuclear migration" for further consideration by *eLife*. Your revised article has been evaluated by Piali Sengupta (Senior Editor) and a Reviewing Editor.

The manuscript has been improved with the addition of new data and analysis further exploring ARL3-Y90C's action as a dominant negative through strong Arl13b binding. These new experiments show that sequestration of ARL13B by this mutant contributes to the nuclear migration defects. Thus, ARL13B-90C is clearly not simply a rapid cycling mutant and the nuclear migration phenotype is contributed by multiple factors. However, the manuscript in its present form does not reflect the nuanced nature of this mutant. Please note that other important issues remain in the current manuscript that need to be addressed, as outlined below:

First, it would help to orient the readers on the role of ARL-trafficked ciliary proteins with respect to neuronal migration, a relatively new phenotype that you study. The current addition on nuclear migration in the introduction is helpful, but please mention the limitations of our current knowledge in this regard, and why you are unable to test for defects in ciliary enrichment of lipidated proteins in your mutants that contribute to the final disease.

Second, the new experiments using co-expression of ARL13B partially restoring the ARL3-Y90C phenotype suggest that the "the migration defect is caused at least in part by Arl3-Y90C acting as a dominant negative (Line 175)". Going by comments from all the reviewers, and our reading of the manuscript, it is likely that this experiment along with the data from the Y90C/R149H active mutant (which does not strongly bind to ARl13B) suggest that the ARL13B sequestering activity of ARL3-Y90C mutant does contribute to the examined phenotype. The dual nature of this mutant should be reflected in the text and abstract.

Third, multiple sections of the manuscript including the title and abstract posit a functional role for a "ciliary gradient" for ARL3. However, the argument for such a gradient needs clarity (eg in lines 350 and 355) and without sufficient explanation, are open to other interpretations.

Finally, please also provide quantification of IPs in Figure 3A and number of experiments performed in biochemical experiments throughout the text. Figure 6-Supp 1: please define asterisks. Please explain the prominent band in middle lane of the Flag IB.

---

## [Author Response]

Essential revisions:As you will see that all the reviewers agree that the paper is of interest to cell biologists studying ciliogenesis and specifically vertebrate photoreceptors. Overall, they mention that the experiments are properly controlled, executed, and analyzed. The development of a method to analyze snapshots of the interaction between ARL3 and its interactors is also a strength of the paper. However, significant concerns have been raised by all the reviewers. We highlight two major concerns at the heart of the paper's current message that need to be addressed before publication.1. How does nuclear migration failure impact ciliogenesis in the outer segment? The reviewers suggest that further analysis of the trafficking defects of effectors and cilium length for electroporated variants could better link cilium/trafficking defects with the nuclei migration phenotypes.

Previous literature suggests that rod outer segment formation is unaffected by nuclear migration defects, which is in line with our own observations. Electroporation of Arl3 mutant constructs appeared to have no effect on outer segment formation or lipidated protein trafficking to the outer segment in adult rods (more details provided in comments below). We agree that this result seems counterintuitive to our hypothesis that the nuclear migration defect observed in rods expressing Arl3 mutants is likely caused by a ciliary trafficking defect. The problem is that we performed all our analysis in mature retinas, but nuclear migration is a developmental process. An in-depth analysis would need to be done to identify how Arl3 mutants affect ciliary trafficking in developing rods at peak migration, which is beyond the scope of this paper. Since the developmental timing of nuclear migration is important for our manuscript, we have added additional introduction to clarify. In mature retinas the rod cilium, aka the outer segment, is fully formed and filled with ~800-1000 flattened ‘disc’ membranes, so cilium length is a product of ongoing disc turnover: new discs are built at the base and old discs shed from the tip. Outer segment lengths can vary across the retina (central to peripheral) and can be affected by retinal detachment during collection or by timing of when eyes were collected (especially at P21 when outer segment are still growing), so we do not feel the electroporation method provides for accurate measurements. Using semi-thin retinal cross-sections, a previous study showed that outer segment length was unaffected in a transgenic model of Arl3-Q71L that has a pronounced nuclear migration defect, so we do not expect any alterations in outer segment length in our electroporated models.

2. Testing alternative hypotheses for the Y90C mutant with regards to (a) sequestration of the GEF ARL13B in preventing activation of the endogenous ARL3, and (b) effects on subcellular localization. The reviewers suggest that these alternative models might be tested by the rescue of ARL3 Y90C driven phenotype upon co-expression of ARL13B and detailed imaging and analysis for the ARL3 variants/potential effectors in photoreceptor subcellular segments.

These are excellent suggestions made by the reviewers and we have included new experiments (Figure 3, Figure 3-SupFigure 1, and Figure 3-SupFigure 2) addressing Arl3-Y90C binding to Arl13B. Our new experimental data shows that co-expressing Arl3-Y90C-FLAG and Arl13B-myc rescues the nuclear migration defect. This suggests that increasing the available pool of Arl13B restores activation of Arl3, which further supports our hypothesis that Arl3-Y90C acts as a dominant negative by blocking activation of endogenous Arl3. We also include supplemental data showing that endogenous Arl13B and overexpressed Arl13B-myc localization is not altered by Arl3-Y90C expression (Figure 3-SupFigure 1-2), confirming that Arl3-Y90C blocks Arl13B activity at the cilium and not through sequestering Arl13B away from the cilium.

Reviewer #1 (Recommendations for the authors):1) Figure 1: It would be better to have a description of what the value means on the left side as well as the bottom side of the scatter plot. The same is true for the histogram. For the histogram, the labels that indicate which mutant the authors expressed should go to the right side of the histogram, rather than the left side, as the labels are at the place where the X-axis of the histogram is located.Also, what is the unit of the y-axis of the histogram? Is this indicating the number? Or percentage?

We have included labels on the scatter plots and histograms. Y-axis of scatter plots: Individual FLAG Nuclei Normalized to ONL. Label above the histograms: Binned FLAG Nuclei Position (%).

2) Figure 2D: The figure legend indicated that the scale bar is 5µm, but there is no scale bar in the image.

Scale bar has been added.

3) Figure 3E: Why there are so many dots here? Are these mixture of the normalized signal intensity of the bands corresponding to UNC119 and BART? If this comes from multiple experiments, how many experiments did the author perform? Are the experiments technical replicates? Or biological replicates? Why does the number of dots differ among the samples?In Figure 3D, the signal intensity of the 45kDa and 55kDa in D129N cells does not look as strong as the bands in Y90C cells. However, the quantification shown in Figure 3E indicated that the signal intensity of the bands in D129N cells is much stronger than those of Y90C cells. Is this because the authors normalized the intensity by the intensity of the FLAG-band?

Yes, UNC119 and BART bands are normalized to the intensity of the FLAG-band. More precise wording has been added to the *Statistical analysis* section of the methods to explain the normalization. The bands analyzed were a part of 7 separate experiments; every experiment included one T31N lane and one Q71L lane, but some experiments included multiple lanes of other mutants. Each lane is the result of a separate transfection of a 10cm plate of 293 cells.

Arl3-D129N-FLAG expression in AD293T cells was consistently lower regardless of the amount of plasmid DNA that was transfected. This resulted in less uncrosslinked Arl3-D129N-FLAG, around the same amount of absolute Arl3-D129N-FLAG crosslinked to UNC119 and BART as Arl3-Y90C-FLAG, and overall less background in the lane. Due to the confounding variable of lower expression, we did not perform statistics comparing D129N to Q71L and Y90C. Since it is easy to appreciate that D129N is forming active complexes in the blots, so included D67V the on the graph.

4) Figure 3F, Figure 3G, and Figure 4A: It would be great if the authors could have quantification just like the authors did in Figure 3E.

No quantification was performed because the bands are either present or absent, language was added regarding the number of biological replicates.

5) Line59: R991 should be R99I.

Done.

6) Line 189: "It was unable to form complexes with expressed with ARL13B-GFP".Should this be "It was unable to form complexes when expressed with ARL13B-GFP"?

Yes, fixed.

7) Line 283-284: "but appears to subtly change the binding properties toward the effectors PDEδ and UNC119." What is this sentence referring to? Figure 2A? or Figure 3C? The affinity towards PDEδ seems to be similar between Q71L and D67V in Figure 2A?

This is taken from the discussion where we go on to clarify and provide experimental evidence: “Even though Arl3-D67V can bind PDEδ and RP2, it does not bind them both at the same time, a Arl3-Q71L feature shown both here and previously (Figure 2-SupFigure 2; Veltel et al., 2008b). We also found that Arl3-D67V disrupted UNC119 binding (Figure 4-SupFigure 1). This effector specificity is surprising considering how similar the binding interfaces are in the UNC119a-Arl2/3 and PDEδ-Arl2 crystal structures (Hanzal-Bayer et al., 2002; Ismail et al., 2012).”

8) Line 353: I might have missed it, but I did not find what expression vectors did authors use for the injection to the eye. Is the backbone of the expression vector pRho-DsRed? (as indicated in line 356)? Or authors just took the promoter out of the pRho-DsRed? In this case, what is the backbone of the expression vector?

The backbone is from the pRho-DsRed plasmid.

9) Line 440: Where is the GTP coming from? Is the catalog number lacking? Also, many other chemicals lack the catalog number. It would be better to include those.

Now included in methods.

10) Line 439-441: Wording is a little confusing. Maybe "10mM EDTA in the presence or the absence of 10mM GTP, or 65mM MgCl2 in the presence of 10mM GTP"?

Thank you updated.

11) Line 465: PBST. Is this PBS containing 0.1% Tween-20?

Yes, included in methods.

12) 477: 37oC. the unit (degree Celsius) is not displayed correctly.

Fixed.

13) Line 497-501: It is not very clear how the authors measured the intensity. Did the author use Li-COR software? Or Image J? Did the author subtract the background?

Band intensity was measured using Li-COR software. See statement in Methods, “Bands were visualized and quantified using the Odyssey CLx infrared imaging system (LiCor Bioscience).”

Reviewer #2 (Recommendations for the authors):1. The authors never examine actual ciliogenesis in their retinal model – nuclear migration is the only readout. First, some description of how nuclear migration impacts retinal development should be included in the introduction. More importantly, however, the authors should specifically show the morphology of the cilia in cells expressing the various Arl3 mutants. This is particularly important for experiments in which the expression of effectors or cargo relieves the nuclear migration phenotype induced by the mutants. Do effectors and/or cargo reach the cilia, and are they distributed normally, even if nuclear migration is rescued? Immunostaining for PDEd, UNC199A, INPP5E, and NPHP3 would be essential for this purpose.

The effect Arl3 dominant mutations have on trafficking of lipidated cargos to the outer segment is an interesting question. A previous papers showed that Arl3-Q71L overexpressing rods did not cause major changes in lipidated protein mislocalization, as originally reported in Arl3 knockout mice. Instead, transgenic Q71L mice only showed mislocalization of PDE. We attempted to investigate lipidated protein localization in our electroporated models; however, we never observed mislocalization of ciliary or outer segment lipidated cargos (i.e. GRK1, transducin, Rab28, and PDE) in wild type mature rods that were overexpressing Arl3 mutants, even with Q71L. Since our method did not replicate the previous findings using transgenic mice, we were left to move forward with the most reproducible phenotype… defects in nuclear migration. We are now interested in identifying alterations in ciliary transport during this developmental process and hope to uncover the mechanism in future studies.

2. The authors should expand their discussion of the Y90C mutant to include its action as a dominant-negative, and the complex nature of its features. It is clearly not simply a rapid cycling mutant.

More experiments addressing Y90C binding to Arl13B have been added.

3. Do any of these mutants have phenotypes in non-retinal ciliated cells? If this is known it could simply be described, but if not they could be tested simply in cell culture models.

Human patients with D67V and Y90C have non-syndromic retinal degeneration suggesting that the phenotype is retina specific.

Reviewer #3 (Recommendations for the authors):– The word "gradient" is stated throughout the text, including in the title. However, none of the experiments ever directly assess this. And is it really a gradient? Or just subcellular compartmentalization segregating the specialized cilium of photoreceptors and the cell body?

We are borrowing the terminology first coined regarding the difference in the concentration of Ran-GTP found within the nucleus versus outside the nucleus. Although our experiments do not directly measure the magnitude or relative steepness of the Arl3-GTP difference across the ciliary entrance, they do indirectly show this phenomenon.

– Since spatial localization of Arl3, Arl13B, RP2, and Arl3 effectors PDEd and UNC119, seem to be key in the overall model of differential subcellular activation of Arl3, it should be better highlighted in the introduction where these proteins are localized in photoreceptors specifically, with the corresponding original references.

Introduction has been edited to include more details.

– Line 69 (introduction): it is stated that "rod nuclear displacement has also been observed in RP2 KO mouse" with direct references to "Li et al., 2013 and Mookherjee et al., 2015". However, rod nuclei displacement was never directly assessed on these papers. Indeed, there seems to be a phenotype in ONL/OPL/INL of RP2 KO mice, which could potentially be displacement of rod nuclei. This is interesting and should be actually evaluated. I suggest an experiment to directly assess this in this mice KO line or by knocking-down RP2 using in a similar strategy to the one the authors use to deliver transgenes to P0 mice retinas by means of electroporation.

This is a great experimental aim that we plan to follow up on in the future, but it is outside the current scope of this work.

– Line 75 (intro): "using in vivo mouse models…". The word "models" should be substituted by "experiments".

Done.

– Line 76 (intro): "show that this aberrant activity disrupts the Arl3-GTP gradient in the cilium". This is actually never directly shown in the manuscript. Combination of multiple experiments together with differential subcellular protein localization, suggest, and suggest only, that this is the case. The wording should be adjusted.

We kindly disagree. We show that dominant mutations alter the levels of Arl3-GTP independent of Arl13B, thereby, increasing activity outside the cilium. Further, we perform multiple experiments showing that increasing Arl3-GTP in the cilium or reducing Arl3-GTP outside the cilium rescues the phenotype. Together, it is fair to say that aberrant activity disrupts the Arl3-GTP ciliary gradient.

– Line 77 (intro): the rescue experiments were only performed for the Y90C variant, not for the D67V. This should be clear during the text. The way it's written implies that rescue experiments were done for both of the mutations.

Done.

– Line 84 (intro): "…in rods using…" modify to "…in mice rods using…"

Done.

– More detailed imaging and analysis should be performed for the specialized cilium of photoreceptors (inner segment, connecting cilium and outer segment). Specifically show in more detail the FLAG-staining for each variant. Even though it is stated in the text that "while both Arl3-D67V-FLAG and Arl3-Y90C-FLAG expression and localization were indistinguishable from wild-type Arl3-FLAG", some images do not represent this. For instance, Y90C-FLAG signal seems to be mostly staining outer segments, while D67V seems to mostly stain inner segments, at least in the representative images shown. Is this always the case?

We have changed the text to better represent the images shown. We appreciate the thorough analysis the reviewer made regarding subtle differences in the FLAG pattern.

– Analysis should be done to measure cilium length for each condition and also between INL-displaced, basally-displaced and non-displaced rods electroporated with the different variants. In addition, it should be evaluated if there is a correlation between cilium length and rod displacement. This is important in linking both cilium and nuclei, the main point of the manuscript.

As mentioned, rod nuclear migration occurs prior to outer segment formation. All our images are taken at the P21 timepoint when the outer segment is mature and we feel it is outside the scope of this paper to study developmental aspects of this process. Further, variabilities between each electroporated patch (location, timing, retinal detachment, etc) prevents accurate measurement of outer segment length.

– "skew" values are a valuable and sophisticated quantification measure of the phenotype and should be added to Figure 1 (and to all other figures, when applicable) to highlight the phenotype.

We feel that we are showing the phenotype multiple ways in each figure: representative images, histogram, scatter plots. We prefer to keep it simple and leave the skew value to Table 1.

– In addition to the skewness analysis, the number of displaced rods normalized to total number of electroporated ones should also be quantified for each condition to better compare and evaluate the phenotype (this is somewhat already shown by the histogram, but adding a percent count will solidify the phenotype finding). This should be done for Figure 1 and all other figures, when applicable. In addition, these percentages should be discussed in the paper. A related question that should be discussed is, since these are dominant variants why isn't the phenotype more dramatic?

The percent of nuclei displaced to the INL has now been included in the manuscript and table. We feel that the migration phenotype is very dramatic and nicely appreciated by our electroporation method that allows for mutant rods to be easily distinguished within the wild type ONL.

– Variant D67V is not as well characterized as Y90C, even though it is introduced with the same significance as the Y90C. What's the model for how D67V is leading to rod displacement, since it does not bind RP2 like the Q71L? Can you add a schematic for this variant to Figure 3H.

The D67V human mutation was published in 2021, after many of our experiments testing Y90C and Q71L were completed. Since this was the second dominant human mutation identified in Arl3, we felt it was important to see if it also produced a nuclear migration defect – and it did! So, we tested Arl3-D67V GTP binding and found it behaves as a constitutively active mutant, like Q71L. But as the reviewer points out has minor differences in its effector binding.

We found that D67V DOES bind to RP2 (see Figure 4C), but not when in a complex with PDEδ. We also found that D67V does not appear to bind to UNC119A. These differences in effector binding are likely due the D67V mutation being in the switch 2 region of ARL3, which will directly impact its binding to PDEδ and UNC119. We have added D67V to the cartoon in Figure 4H as well as added text in the manuscript clarifying this point.

– According to your model, Q71L variant is responsible for rod nuclei displacement due to constant binding to RP2, sequestering it and preventing it to inactivate endogenous Arl3 at the base of the cilium. A straight-forward experiment to test this model is to perform a rescue experiment by co-expression of the variant together with RP2, similar to what was done with co-overexpression of UNC119 or PDE δ. In addition, the Q71L [noRP2] triple mutant should also rescue the rod nuclei phenotype, according to your model.

Yes, this is a great idea and we attempted to express both Arl3-E164A/D168A double mutant that does not bind RP2 and Arl3-Q71L/E164A/D168A triple mutant; however, we never saw FLAG expression in the rods at P21. Interestingly, by using the soluble transfection marker to identify transfected rods we did see a migration defect with the Arl3-E164A/D168A mutant when driven by the high-expression pRho promoter but not the lower-expression pRK promoter—the same pattern seen when we attempted to rescue Arl3-Q71L with the chaperones. This suggests that disrupting binding of active Arl3 to RP2 to allow for wildtype inactivation can rescue the phenotype as long as the amount of Arl3-E164A/D168A is not above a certain threshold, but since we could not definitively show FLAG expression in these rods, we felt we could not include the data.

– Perform another rescue experiment of Arl3Y90C driven phenotype by co-overexpression of Arl13B. Increasing the levels of Arl13B should also rescue the rod displacement if there is enough Arl13B to activate endogenous Arl3, according to your proposed model.

This was an excellent idea. We have completed the rescue experiments where we co-electroporated Arl3-Y90C-FLAG with Arl13B-myc and found that increasing the available pool of Arl13B rescued the migration defect. We also show that Arl13B-myc expression remains predominantly in outer segment compartment, even with Arl3-Y90C, further suggesting that Arl3-Y90C’s dominant negative function is to bind onto Arl13B blocking activation of endogenous Arl3 and not through removing Arl13B from the cilium.

– To meet the human relevance justifying your studies (that is, Arl3 variants associated with dominant inherited retinal degeneration), assess potential degenerative phenotype in older mice, such as 2-months old (when severe retinal degeneration is observed in other mouse models, such as Arl3-Q71L transgenic mouse). This experiment would also address a simple potential delay in rod migration, although very unlikely.

The electroporation technique we use has expression only in a subset of rod photoreceptors, so is not a good method for assessing retinal degeneration outcomes.

– Discuss why the Q71L variant has not been associated with human disease, either retina-specific or more generic ciliopathies.

While an interesting question, we honestly have no idea and would need to defer to a human geneticist to explain the rate at which spontaneous human mutations arise in the population. We are not comfortable adding any discussion about this, as we are not experts.

– Figure 3C: there is an extra band , highlighted by a red arrowhead. This band is mentioned in the figure legend but never in the Results section. Please comment on it.

Done.

– Figure 4B: in the rescue experiment of Arl3 T31N/Y90C: do you see a degeneration phenotype if you look at older mice, similar to Arl3KO?

See response above, our electroporation technique limits ability to study degeneration.

– Consider reorganizing your table by listing the variants by the order they appear in the main text. Consider adding an extra column for "binding effectors: PDEd and UNC119" and state if they bind the specific variant. This table is a great reference.

Adding!

– Comment should be made as to inconsistencies regarding using human or mouse sequences (for instance, human PDEd, but mouse UNC119). An explanation should be given, such as high level of protein conservation.

We have included more details about the species used, however, it is common practice in the small GTPase field, and specifically for Arl3, to use species interchangeably. Further, we ensure to control for these variables by using wild type, Q71L and T31N mutants of the same species throughout the manuscript.

– State the ratio of each plasmid when co-expressing multiple proteins.

Done.

Figure 5: the beginning of the figure legend "Arl3 driven nuclear…" should be "Arl3-Y90C driven nuclear…".

Done.

Figure 5 SpFigure 1: describe red asterisks.

Done.

[Editors' note: further revisions were suggested prior to acceptance, as described below.]

The manuscript has been improved with the addition of new data and analysis further exploring ARL3-Y90C's action as a dominant negative through strong Arl13b binding. These new experiments show that sequestration of ARL13B by this mutant contributes to the nuclear migration defects. Thus, ARL13B-90C is clearly not simply a rapid cycling mutant and the nuclear migration phenotype is contributed by multiple factors. However, the manuscript in its present form does not reflect the nuanced nature of this mutant. Please note that other important issues remain in the current manuscript that need to be addressed, as outlined below:First, it would help to orient the readers on the role of ARL-trafficked ciliary proteins with respect to neuronal migration, a relatively new phenotype that you study. The current addition on nuclear migration in the introduction is helpful, but please mention the limitations of our current knowledge in this regard, and why you are unable to test for defects in ciliary enrichment of lipidated proteins in your mutants that contribute to the final disease.

We have added additional information to throughout our manuscript to clarify that the defect occurs during development, when rods are “immature”. We also added more to discussion to explaining limitations and future experiments to identify the Arl3 cargo responsible for nuclear migration.

Second, the new experiments using co-expression of ARL13B partially restoring the ARL3-Y90C phenotype suggest that the "the migration defect is caused at least in part by Arl3-Y90C acting as a dominant negative (Line 175)". Going by comments from all the reviewers, and our reading of the manuscript, it is likely that this experiment along with the data from the Y90C/R149H active mutant (which does not strongly bind to ARl13B) suggest that the ARL13B sequestering activity of ARL3-Y90C mutant does contribute to the examined phenotype. The dual nature of this mutant should be reflected in the text and abstract.

Yes! We are also very excited by the new data showing Arl13B overexpression can rescue the Arl3-Y90C migration phenotype. Considering this data, we altered the text to emphasize the dominant negative feature of this mutation. However, we feel strongly that introducing the Y90C mutation as fast cycling is very important as one outcome of reduced nucleotide binding is increased Arl13B binding (shown in Figure 3A). In the small GTPase field, it is well known that fast cycling mutations increase GEF affinity, and these mutations are often used to identify the GEF for a particular small GTPase (Gotthardt et al. *eLife* 2015; Berken et al., 2005; Cool et al., 1999). Furthermore, we show in Figure 5 that the Arl3-Y90C/T31N mutation, which maintains Arl13B binding (Figure 3A) rescues the migration phenotype. This means that dominant negative Arl3 binding to Arl13B is not sufficient to cause the defect.

Third, multiple sections of the manuscript including the title and abstract posit a functional role for a "ciliary gradient" for ARL3. However, the argument for such a gradient needs clarity (eg in lines 350 and 355) and without sufficient explanation, are open to other interpretations.

Additional information is provided in the discussion to help clarify experiments showing that an reactivating Arl3-GTP in the cilia is required for proper nuclear migration in rods.

Finally, please also provide quantification of IPs in Figure 3A and number of experiments performed in biochemical experiments throughout the text. Figure 6-Supp 1: please define asterisks. Please explain the prominent band in middle lane of the Flag IB.

Figure 3A shows representative IPs between Arl3-FLAG constructs and Arl13B-GFP. In the text, we interpret this data as either binding or not binding, since IPs do not assess affinity. We have now added a graph showing amount of Arl13B-GFP bound to Arl3-FLAG in Figure 3-SupFigure 3 for all the replicates for each construct. Additional information has been added to methods describing number of replicates and statistics.

For Figure 6-SupFigure 1, we have now added labels for each lane of the Western blot as well as additional information in the figure legend to help clarify. The red asterisks mark crosslinked Arl3-Y90C complexes that are consistently identified in the in vivo crosslinking experiments initially shown in Figure 4. The top asterisk indicates the Arl3-Y90C/UNC119 complex and overexpression of UNC119-myc increases the presence of this band (center lane). Importantly, we see that the other crosslinked Arl3-Y90C complex (presumably with BART) is reduced when Arl3-Y90C is bound to UNC119c. The same reduction occurs for both Arl3-Y90C effector complexes when Arl13B-GFP is expressed. This assay was used to see how overexpression of Arl3 effectors (UNC119 and Arl13B) affects the level of active Arl3-Y90C in the cell. We found activity was reduced.